# The UrbEm Hybrid Method to Derive High-Resolution Emissions for City-Scale Air Quality Modeling

Martin Otto Paul Ramacher [1], Anastasia Kakouri [2,3], Orestis Speyer [2], Josefine Feldner [1], Matthias Karl [1], Renske Timmermans [4], Hugo Denier van der Gon [4], Jeroen Kuenen [4], Evangelos Gerasopoulos [2] and Eleni Athanasopoulou [2,*]

1 Helmholtz-Zentrum Hereon, Max-Planck-Str. 1, D-21502 Geesthacht, Germany; martin.ramacher@hereon.de (M.O.P.R.); josefine.feldner@hereon.de (J.F.); matthias.karl@hereon.de (M.K.)
2 Institute for Environmental Research and Sustainable Development, National Observatory of Athens, 11810 Athens, Greece; nkakouri@noa.gr (A.K.); ospeyer@noa.gr (O.S.); egera@noa.gr (E.G.)
3 Department of the Environment, University of the Aegean, 81100 Mytilene, Greece
4 TNO, Department of Climate, Air and Sustainability, Princetonlaan 6, 3584 CB Utrecht, The Netherlands; renske.timmermans@tno.nl (R.T.); hugo.deniervandergon@tno.nl (H.D.v.d.G.); jeroen.kuenen@tno.nl (J.K.)
* Correspondence: eathana@noa.gr

**Abstract:** As cities are growing in size and complexity, the estimation of air pollution exposure requires a detailed spatial representation of air pollution levels, rather than homogenous fields, provided by global- or regional-scale models. A critical input for city-scale modeling is a timely and spatially resolved emission inventory. Bottom–up approaches to create urban-scale emission inventories can be a demanding and time-consuming task, whereas local emission rates derived from a top–down approach may lack accuracy. In the frame of this study, the UrbEm approach of downscaling gridded emission inventories is developed, investing upon existing, open access, and credible emission data sources. As a proof-of-concept, the regional anthropogenic emissions by Copernicus Atmospheric Monitoring Service (CAMS) are handled with a top–down approach, creating an added-value product of anthropogenic emissions of trace gases and particulate matter for any city (or area) of Europe, at the desired spatial resolution down to 1 km. The disaggregation is based on contemporary proxies for the European area (e.g., Global Human Settlement population data, Urban Atlas 2012, Corine, OpenStreetMap data). The UrbEm approach is realized as a fully automated software tool to produce a detailed mapping of industrial (point), (road-) transport (line), and residential/agricultural/other (area) emission sources. Line sources are of particular value for air quality studies at the urban scale, as they enable explicit treatment of line sources by models capturing among others the street canyon effect and offer an overall better representation of the critical road transport sector. The UrbEm approach is an efficient solution for such studies and constitutes a fully credible option in case high-resolution emission inventories do not exist for a city (or area) of interest. The validity of UrbEm is examined through the evaluation of high-resolution air pollution predictions over Athens and Hamburg against in situ measurements. In addition to a better spatial representation of emission sources and especially hotspots, the air quality modeling results show that UrbEm outputs, when compared to a uniform spatial disaggregation, have an impact on $NO_2$ predictions up to 70% for urban regions with complex topographies, which corresponds to a big improvement of model accuracy (FAC2 > 0.5), especially at the source-impacted sites.

**Keywords:** air pollution modeling; air quality modeling; emission rate modeling; urban air pollution; chemistry transport modeling; EPISODE-CityChem

## 1. Introduction

The percentage of the population residing in urban areas in Europe continues to increase from 74.9% in 2019; it is expected to reach 77.5% (83.7%) by 2030 (2050) [1,2]. In turn, this will increase the urban population exposed to air pollutants and the consequent

health impacts. Presently, despite past reductions in emissions that have taken place in most European countries, a significant proportion of the urban population in the EU-28 is still exposed to concentrations of certain air pollutants above the EU limit values. This is even more so when the more stringent WHO air quality guideline values are taken into account (i.e., the percentage population exposure to $PM_{2.5}$ above limit values rises from 6–8% (EU limit) to 74–81% (WHO limit). With respect to $PM_{10}$, the respective exceedances were 13–19% (42–52%), for surface $O_3$, they were 12–29% (95–98%), and for $NO_2$, they were 7–8% both for the EU and WHO [3]. The estimated premature deaths in the 27 EU Member States and the United Kingdom attributed to $PM_{2.5}$, $NO_2$, and $O_3$ exposure are 374,000, 68,000, and 14,000, respectively, while the number of years of life lost (YLL) per 100,000 inhabitants is estimated at 800, 100, and 30 [3]. The ever-rising amount of evidence regarding the negative effects of air pollution on health—evidence assessed as having high or moderate certainty of an association between a pollutant and a specific health outcome—has led to new and even more stringent guidelines set by the WHO in September 2021 [4].

As cities are growing in size and feature inherent complexity, urban air quality management requires a more detailed spatial representation of air pollution levels than given by (a) the available number of stationary measurement stations and (b) the current resolution (one to tens of kilometers) provided by regional Eulerian grid air pollution models [5]. These models, inter alia, fail to capture concentration gradients that typically occur near heavily trafficked streets [6], with further implications on exposure [7] and the choice of mitigation measures. The accuracy of the assessment of the effectiveness of these measures is dependent on the robustness of the emission (the flux of certain trace gases and particles into the atmosphere) inventories and their efficacy for scenario analysis to assess measures for emission reductions and improved air quality [8,9]. Therefore, emission inventories are considered a key component in an urban air quality management plan [10]. Emission inventories come at various scales, from global to regional and local. For the purposes of this study, we will only consider the latter two and, in particular, how well they serve the urban domain, as it is there where urban health studies are often hindered by the lack of city-scale emission inventories.

Several regional emission inventories exist in Europe, with spatial resolution usually between 7 and 11 km. A spatial inter-comparison of some of these top–down and proxy-based inventories, including EDGAR v4.3.1, TNO-MACCIII, EMEP, and JRC07, has been performed in representative European urban areas [11], and sources herein. The authors acknowledge that although bottom–up inventories do often exist for major cities, providing more accurate information at a higher spatial resolution is still of utmost importance to be able to rely on consistent and harmonized European-wide inventories for extensive air quality modeling. However, the study also identified significant differences between regional emission inventories in these cities due to the choices made in terms of the disaggregation approach in the industrial sector, where the use of the population density as a proxy for the diffuse fraction results in an over-allocation of emissions in urban areas and in the residential sector, where spatial patterns and variation of emissions from wood and coal burning are not captured.

As noted in Kadaverugu et al. [12], CTMs require emission inventories representative of the study area. Similarly, appropriate spatial and temporal resolution inventories on a grid that is compatible with city-scale models is imperative. Currently, many countries provide emission inventories down to $1 \times 1$ $km^2$ [10], although no universal standards are available; thus, applications rely on urban or local-scale project initiatives [5]. The Forum for Air Quality Modeling (FAIRMODE) has recently published a report providing recommendations to build emissions for fine-scale applications [13]. This report advocates for an expansion of the existing EMEP/EEA [14] emission guidance document to ensure that spatial disaggregation approaches provide appropriate results for local/urban emission assessments, for an adoption of the Gridded Nomenclature For Reporting (GNFR) and for national efforts toward urban-scale emission compilation following bottom–up approaches.

A bottom–up approach relies mostly on local activity estimates collected over the area of interest (e.g., traffic counts for road segments in a city). One recent example, albeit more generalized, was the approach followed by Guevara et al. [15] where, in sync with EMEP guidelines, a method for estimating and preparing high-resolution bottom–up emissions from multiple anthropogenic sources in a flexible and transparent way was developed. The approach presents a Europe-wide applicability as long as the required input data are available. A top–down approach, on the other hand, distributes emission totals spatially (country or region, e.g., derived from total fuel sales) according to gridded proxies (e.g., population, land use). Nevertheless, bottom–up approaches have usually different levels of detail or granularity, as not all parameters needed for a pure bottom–up approach are known and, as a result, bottom–up and top–down modeling approaches are often combined [10].

Given the fact that bottom–up inventories are resource intensive and are not built in a standardized manner [16], methodological homogeneity between different cities and overall compliance with AQ Directives is not easily achieved. Moreover, bottom–up approaches are in general site-specific and resource intensive, so the support for their compilation in a consistent basis does not exist for every city. For this, we developed a hybrid and modular approach to enable the construction of the desired high-resolution emission fields. The approach is hybrid, as it commences from downscaling regional emission inventories to a gridded inventory—in this case, the CAMS-REG regional inventories from the Copernicus Atmospheric Monitoring Service (CAMS) [17,18], following a traditional top–down methodology. However, instead of disaggregating solely into areas, it explicitly handles the—critical for the urban environment—line and point sources, so as to accommodate for urban CTM models such as [19–22]. The approach is also modular as it can handle different proxies for spatially disaggregating the regional inventory, in this case JRC's Global Human Settlement Layer (GHSL) [23], Copernicus Land Monitoring Services' Corine Land Cover (CLC) [24], European Pollutant Release and Transfer Register (E-PRTR) point source information (https://industry.eea.europa.eu/, accessed: 23 October 2021), and Open Street Map (OSM) [25] road network data.

This study describes the framework for creating emission inventories for urban air quality simulations across Europe by utilizing generic and publicly available proxies in a consistent manner. While the concept of downscaling coarser emissions, up to the 1 km mark, by utilizing proxies (point, line, or areas) has been visited in the past, such as in the MEGAPOLI project (https://cordis.europa.eu/project/id/212520, accessed: 23 October 2021), where a fine emission inventory was created for the whole of Germany albeit though localized data and effort, here, Europe-wide applicability, consistency, and ease of use were the strategic objectives. The demonstration cities are Hamburg and Athens, i.e., cities with different climatic, geomorphological, and urban growth characteristics, where the accuracy of the produced inventories is evaluated by comparing concentration fields from urban-scale air quality modeling with actual measurements. This framework is completely scalable, sustainable, has a low computational and overall implementation cost, and is of high and critical importance for air quality modelers, especially when national/local emission data are either unavailable or outdated.

The development of the city-scale emission inventories is based on a hybrid approach, investing upon existing, open access, and credible emission datasets. Their spatial disaggregation is based on contemporary spatial datasets of the European area (Section 2.1). The emission inventories and spatial datasets are processed in the newly developed UrbEm framework (Section 2, Figure 1), which is a hybrid method to derive high-resolution emissions for city-scale air quality modeling. UrbEm is fully automated to produce a detailed mapping of industrial (point), transport (line), residential, agricultural, and other (area) emission sources for any city (or area) of Europe at the desired spatial analysis. These products can be directly used as input for local-scale atmospheric models, for the estimation of air pollution levels within the region of study, at high resolution. In this study, we use emission inventories that were downscaled with the presented approach in the chemistry

transport model EPISODE-CityChem [22] and applied it to two European urban areas, Athens (Greece) and Hamburg (Germany). Finally, the results of the emissions downscaling procedure and their application in EPISODE-CityChem are presented (Section 3.1), evaluated (Section 3.2), and discussed (Section 4).

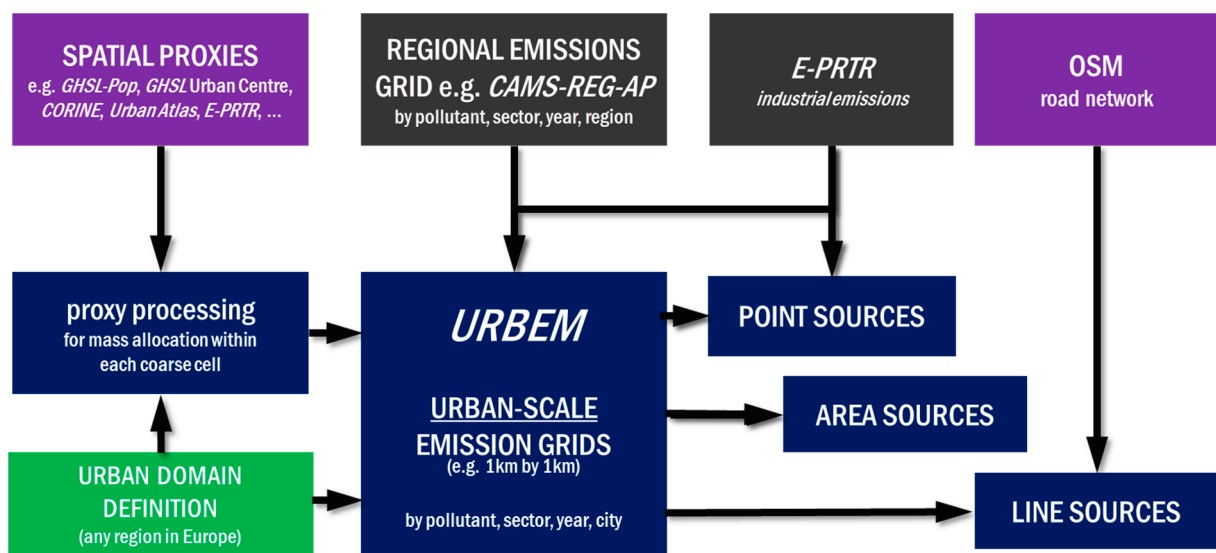

**Figure 1.** General methodology for the Urban Emissions downscaling framework UrbEm.

## 2. The UrbEm Approach for Emissions Downscaling

The developed hybrid approach to arrive at high-resolution area emissions, as well as point and line source emissions for city-scale air quality modeling, follows the generalized framework described in Figure 1. This framework can be applied to any urban region in Europe. Based on an urban domain definition, in line with the requirements of a city-scale air quality model, sector-specific spatial proxies are prepared based on publicly available datasets to distribute CAMS-REG and E-PRTR emission datasets to area, point, and line sources. The overall methodology includes first the general approach to prepare spatial proxies for different sectors (2.1) and second, the application of these proxies in the hybrid-downscaling approach, which is called UrbEm (2.2). A reduced overview on the general methodology is illustrated in Figure 1, while a more detailed insight in the realization of the general methodology is given in Figure 2.

The application of the UrbEm approach is realized as a software tool that is open source distributed via GitHub (https://github.com/martinottopaul/UrbEm, accessed: 23 October 2021). There are two different versions available: (1) based on the interpreted programming language R [26], and (2) a combination of Python (https://www.python.org, accessed: 23 October 2021). Both solutions run on different operating systems on single-core processors and need a minimum of 2 gigabyte RAM. Depending on the extent of the model domain and its resolution, the preparation of proxies, the downscaling of emissions, and the creation of point, area, and line source output files, the required computing time is about 30–45 min (for example, for a 50 km × 50 km domain, with a grid resolution of 1 km × 1 km, including 15,000 line source elements and several point sources).

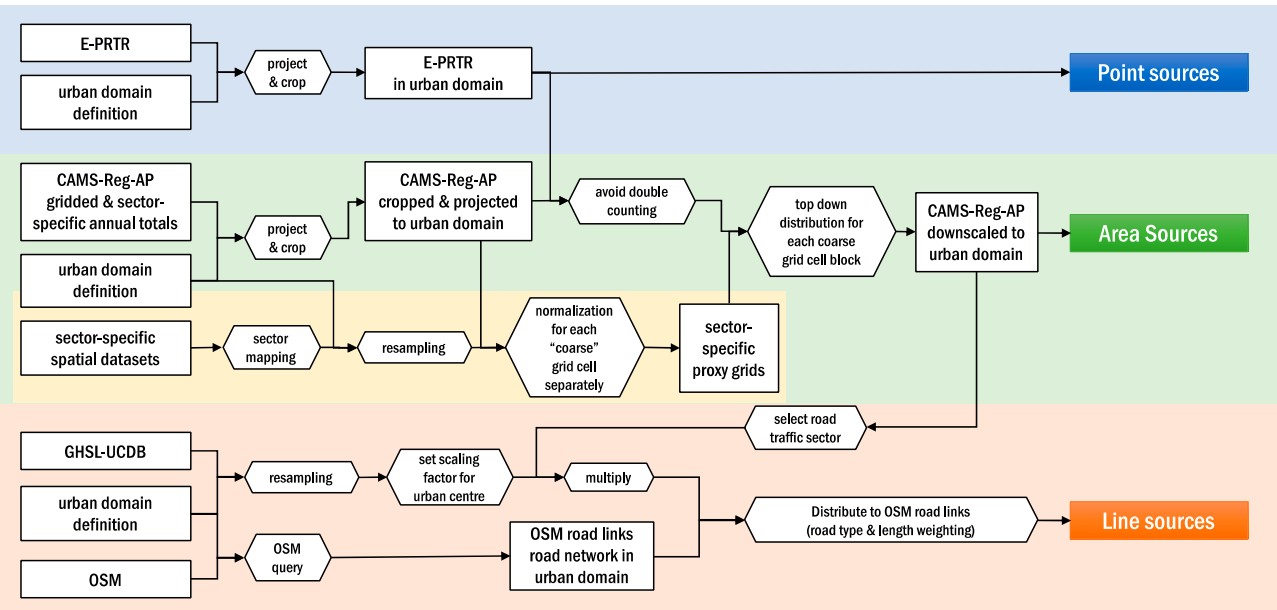

**Figure 2.** Schematic representation of the UrbEm hybrid approach for downscaling regional emission inventories to arrive at urban-scale emission inventories including point, area, and line sources.

### 2.1. Spatial Datasets: Selection and Processing

As illustrated in Figures 1 and 2, the spatial disaggregation of the selected CAMS regional emission inventory CAMS-REG ($\approx 6 \times 6$ km$^2$ grid cells) [17] is performed with sector-specific spatial proxies. These proxies are prepared with publicly available, well-established, contemporary spatial datasets of European (or Global) coverage (for a detailed description of all datasets, see Appendix A). In a first step, suitable spatial datasets are mapped to sectors following SNAP or GNFR nomenclature. Table 1 introduces anthropogenic activities based on SNAP or GNFR classification, which are allocated to the appropriate proxies, such as population density, land type categories (e.g., industrial, agriculture), and road networks.

**Table 1.** The spatial proxies (and their origin) used to disaggregate each anthropogenic activity (expressed as source sectors in SNAP and GNFR) in the proposed downscaling framework. More information on all spatial datasets used can be found in Appendix A.

| Anthropogenic Activity (Source Sector) | Spatial Proxy (Dataset Source) |
| --- | --- |
| Public Power and Refineries (SNAP 1 * or GNFR A) | Polygons hosting Public Power installations (E—PRTR and CLC 2018) combined with Land type characterized as 'Industry' (CLC 2018) |
| Residential Heating (SNAP 2 or GNFR B) | (Residential) population Density (GHS-POP 2015) |
| Fossil Fuel Production and Fugitive (SNAP 5 or GNFR D) | Land type characterized as 'Industry' (CLC 2018) |
| Solvent and Other Use Production (SNAP 6 or GNFR E) | (Residential) population Density (GHS-POP 2015) |
| Road Emissions (SNAP 7: 71,72,73,74,75 or GNFR F) | Major Road Network (OSM) ** consisting of highways, trunks, primary and secondary roads, and their links |
| Non-Road Mobile Emissions (SNAP8): Shipping (GNFR G) | A superposition of Global shipping routes (CIA 2013) and Land type characterized as 'Ports' (CLC 2018) |

**Table 1.** *Cont.*

| Anthropogenic Activity (Source Sector) | Spatial Proxy (Dataset Source) |
|---|---|
| Non-Road Mobile Emissions (SNAP8): Aviation (GNFR H) | Land type characterized as 'Airports' (CLC 2018) |
| Non-Road Mobile Emissions (SNAP8): Off Road Machinery (GNFR I) | Land type characterized as 'Non-Road Mobile Sources' (CLC 2018) relevant to agricultural, industrial, and construction activities |
| Waste Treatment (SNAP 9 or GNFR J) | Polygons hosting waste management installations (E—PRTR and CLC 2018) combined with Land type characterized as 'Agriculture' (CLC 2018) to allocate open waste |
| Agriculture (SNAP 10 or GNFR K and GNFR L) | Land type characterized as 'Agriculture' (CLC 2018) |
| Industrial Combustion and Processes (SNAP 34 or GNFR B) | Polygons hosting installations of mineral or chemical industries and of production (and processing) of wood, paper, metals, animal and vegetable (E—PRTR and CLC 2018) combined with Land type characterized as 'Industry' (CLC 2018) |

* CAMS-REG v.1 (SNAP categorization) uses point sources for the spatial allocation of industrial installations.
** OSM data are periodically (30 days) retrieved to ensure consistency with the frequent official updates of the database.

Population density, as provided within the GHSL-POP layer, is used as a spatial proxy to distribute emissions from residential heating and the use of solvents. The residential population density, mapped for 2015 in 1 km$^2$ spatial resolution, is retrieved from the Global Human Settlement Layer [23].

The CORINE land use/cover (LULC) dataset is used as a spatial proxy for the majority of source sectors (Table 1) [24]. Original LULC gridded data for 2018, with the spatial resolution of 100 m and 44 land types, are aggregated to 1 km$^2$ and reclassified into 12 generalized land types. Reclassified gridded data are either used as spatial proxies for sectors, such as industry and agriculture, or combined with other spatial information in order to create enhanced land use classes.

In particular, reclassified gridded data are combined with global shipping routes [27] and E-PRTR industrial information in order to create new added-value proxies. Global shipping routes for 2013, in vector format, are combined with LULC gridded ports, and a complex proxy for shipping emissions is created. E-PRTR point location information (2019) are combined with the LULC vector Geo-database (2018) and reclassified LULC gridded data. E-PRTR location information, from around 24,000 industrial facilities, divided in 54 sectors and sub-sectors, are reclassified to two general categories, Waste/Wastewater Management and Other Industrial Activities, and these are then spatially joined with the LULC polygons to arrive at a new land use classification. These new land use classes are spatially joined with LULC reclassified gridded data in order to create enhanced proxies for Waste Treatment and Industrial Combustion and Processes emissions.

To further prepare the downscaling methodology of the gridded datasets, the sector-specific proxies are resampled (nearest neighbor), masked by the extent of the selected urban area, and projected to the appropriate coordinate system (e.g., UTM Zones). The resampled sector-specific proxy grids are normalized per "coarse" grid cell (defined by the CAMS-REG resolution) separately, by forming the sum of all resampled "fine" grid cells (defined by the urban domain definition) that are within each "coarse" grid cell and then dividing each grid cell by this sum. Thus, the spatial distribution information of the "coarse" CAMS-REG grid is kept in the sector-specific proxy grids, which follow the urban domain definition.

To account for the road traffic sector, Open Street Map vector data (line shape-files) were applied. We selected eight road types based on highway OSM data to construct the major road network of each urban area (motorway, motorway link, primary, primary link, secondary, secondary link, trunk, trunk link). Thus, for each urban domain, the major road network is represented.

### 2.2. *The Downscaling Method*

The UrbEm approach to downscale the regional emission inventory CAMS-REG—or any coarsely gridded emission inventory—allows for the flexible creation of urban-scale emission inventories which can consist of any combination of area, line, and point source information. Thus, the procedure allows creating emission inventories that are suitable for urban-scale CTM simulations and enable considering the near field dispersion of pollutants by applying specific modules for point and line sources. This framework can be applied to downscale all gridded sector emissions with sectors declared either in SNAP or GNFR nomenclature, depending on the version of the applied CAMS inventory. The mapping of SNAP to GNFR sector nomenclature follows Granier et al. (2019) [18].

### 2.2.1. Point Sources

To create a point source emission inventory, the E-PRTR emissions register is applied to get the annual total emission values per sector and industrial unit. Therefore, the E-PRTR register for the whole of Europe is projected and cropped to a target urban domain definition. Before the point source emission information is written into an output format that can be read by any CTM (e.g., urban-scale EPISODE-CityChem [22]), the annual emission totals for each sector and pollutant from the CAMS-REG emissions are used to cross-check area emission information and avoid double counting (further explained below in Section 2.2.2).

### 2.2.2. Area Sources

To create area source emissions, CAMS-REG regional emissions are combined with spatial datasets that are mapped to be used as sector specific spatial downscaling proxies (Figure 3). Based on the target urban domain's extent, resolution, and projection, the selected CAMS dataset is projected and cropped to the extent of the target urban domain. The result is a grid of annual emission totals with the projection and extent of the target domain but still with the coarse resolution of the regional CAMS emission inventory. In the next step, the emission totals for each sector and pollutant are compared to the sector-specific point source emissions totals, as derived from E-PRTR. If the annual emission total of the sector-specific point source is higher in the same reference year, the annual emission total of the CAMS inventory is dismissed, and only point source information is used. It is here noted that this is a rare case, given the fact that the highly-emitting industrial facilities registered in E-PRTR are low in number (below 100 for 90% of countries); consequently, their fraction situated within the boundaries of any gridded urban domain is even lower. If the annual emission total of the sector-specific point source is lower, the annual emission total of the CAMS inventory is corrected. This is done by dividing the CAMS annual emission total grid by the sum of the annual emission totals of this grid to achieve a normalized grid, containing only information on spatial distribution. Then, this normalized grid will be multiplied with the difference of the annual emission total of the CAMS inventory and the sector-specific point source emissions totals, as derived from E-PRTR. Thus, the spatial information of the CAMS inventory is kept, and double-counting with E-PRTR is avoided. This procedure is mainly necessary for public power and industrial activity sources. In the next step, the corrected grids of annual emission totals are top–down distributed to the normalized grids of sector-specific proxies. Thus, the spatial distribution information of the coarser CAMS grid is considered in downscaling to the high-resolution grid (e.g., $1 \times 1$ km). Nevertheless, caution should be taken with the use of CLC-based spatial proxies for the disaggregation of agricultural emissions, when the focus is on non-urban domains,

including livestock. Due to their high emissions, modern stables should be allocated at points rather than uniformly at the agricultural areas.

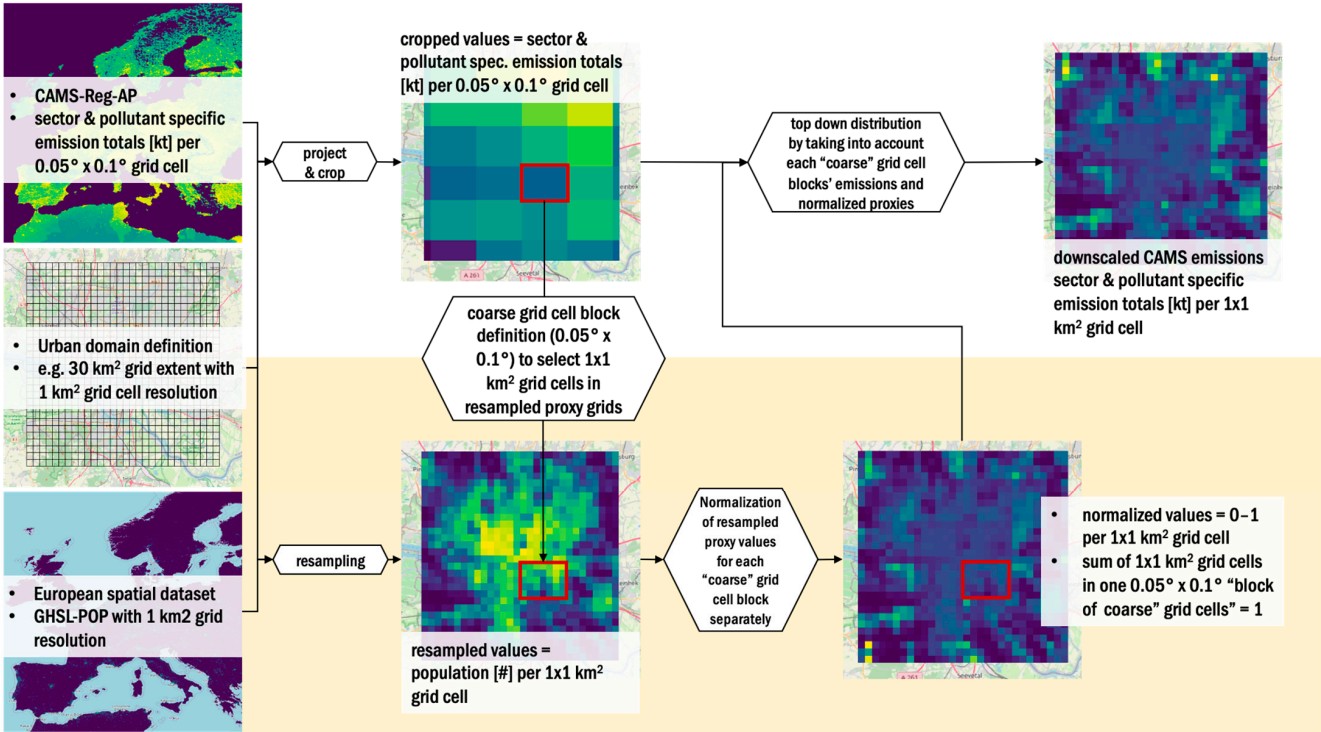

**Figure 3.** Area emissions downscaling procedure to the urban-scale based on an urban domain definition (**top**) and preparation of spatial proxies for downscaling (**bottom**) under consideration of spatial distribution in the "coarse" resolution CAMS-REG (or any other) dataset.

### 2.2.3. Line Sources

To estimate line sources, mainly belonging to the road transport sector, CAMS road transport emissions are firstly downscaled to area sources with the introduced procedure and then distributed to road links derived from OpenStreetMap (OSM) (Figure 4). The transformation of area sources for road transport to line sources generally requires spatial information on road networks as well as information on traffic density or annually averaged daily traffic volumes. Spatial data on traffic densities, vehicle and road types, etc. have been taken into account through investing upon the spatial disaggregation of road transport emissions of the CAMS-REG inventory [17]. As the UrbEm approach aims at minimizing the need for explicit data requirements such as these, the OSM-based methodology was adopted to account for higher traffic densities in urban areas. However, before distributing road traffic area emissions to road links, a factor to alter road traffic emissions in urban centers is applied. This is done to counteract likely underestimations of road traffic emissions in urban areas, when downscaling regional emission inventories to the urban scale, which are frequently recognized as one of the main causes of bias in modeled $NO_X$ concentrations, as demonstrated by Kuik et al. [28]. Kuik et al. investigated the top–down quantification of regional $NO_X$ emissions from traffic in the Berlin area to be applied for urban-scale air quality modeling and estimated "a correction factor for traffic $NO_X$ emissions of ca. 3 [is estimated] for weekday daytime traffic emissions in the core urban area, which corresponds to an overall underestimation of traffic $NO_X$ emissions in the core urban area of ca. 50%" [28] (p 8203). Furthermore, depending on the time and season, the $NO_X$ emission correction factor derived by Kuik et al. ranges between a minimum of ca. 2 (midday in summer) and a maximum of ca. 4.5 (winter morning). Although Kuik et al. derived the averaged emission correction factor of ca. 3 solely for the Berlin urban area

and $NO_X$, we apply this factor as a default factor to increase road traffic emissions for all pollutants in any European urban center before downscaling regional road traffic emissions to the urban scale. This approach was used by Ramacher and Karl [29] in the city of Hamburg and resulted in good agreements with measured values of $NO_X$, $PM_{10}$, and $PM_{2.5}$. Nevertheless, the default correction factor is very likely to vary from city to city and especially for each pollutant, which is why there is a simple option to adjust the factor in the UrbEm approach software application.

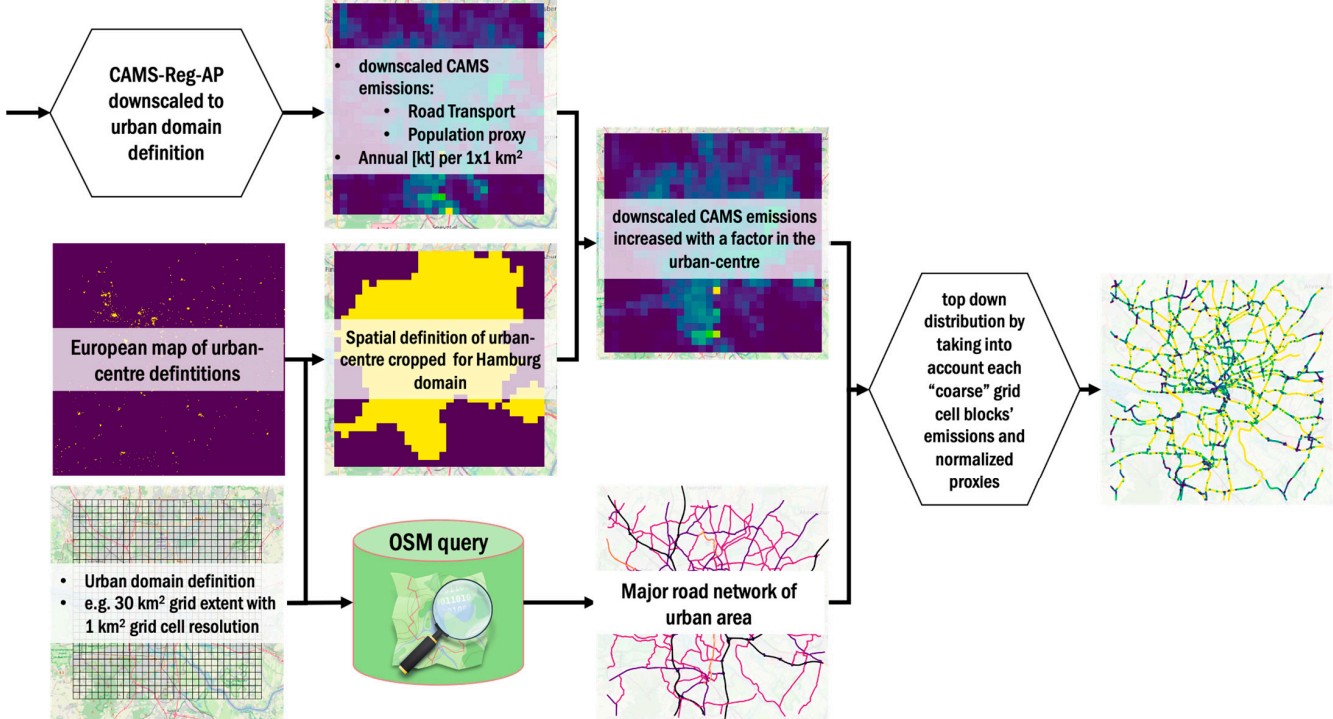

**Figure 4.** Line emissions downscaling procedure in the UrbEm approach.

Based on these findings, the downscaled road traffic area emissions are masked with an Urban Centre Database layer [30] of the Global Human Settlement Layer project (GHSL) of European Commission's Joint Research Centre (Annex IV), and road traffic emissions that are within an area marked as the urban center are multiplied with the default emission correction factor. The road transport emissions modified in this way may be extracted and used as area emissions in case a gridded format is desired; otherwise, they are converted into a dataset of line sources by applying major road types of the OpenStreetMap (OSM) database. Therefore, each grid cell of the downscaled road traffic area emissions is separately intersected with downloaded OSM road links, which are tagged as motorway, trunk, primary, and secondary roads. The intersecting OSM road links' lengths are used to calculate the total road length of all intersecting road links. The total road link length is used to derive a first weighting factor for each intersecting road link. A second weighting factor is derived, based on the different road types of each road link intersecting the grid cell, to account for generic traffic densities of different road types, following the work of Ibarra-Espinosa et al. [31,32]. The combination of both weighting factors allows for top–down distribution of the grid cell emission value to all intersecting road lengths, taking into account length and road type. This is repeated for all grid cells of the road traffic area emissions grid. Thereby, all road traffic area emissions are distributed to OSM road links (line emission sources).

## 3. Methodology Evaluation

### 3.1. Comparison of Air Pollution Emissions

In order to identify the spatial differences and optimization of the European emission inventories (here CAMS datasets) when downscaled at the local level, representative species of selected source sectors are mapped for two scenarios: through the UrbEm approach (UrbEm scenario) and through a uniform spatial disaggregation of the original annual emission rates (CAMS no proxy approach or no-proxy scenario). The two cities that act as demonstrators of the value of the current development are Hamburg (Germany) and Athens (Greece) (Figure 5). Emission totals for the urban domains are compared to official reports or publications (Sections 3.1.1 and 3.1.2), and differences in the spatial distribution provided by the two approaches are discussed. It should be noted that the two sets of gridded emission rates serve also as inputs to the city-scale model applications for both cities (Section 3.2).

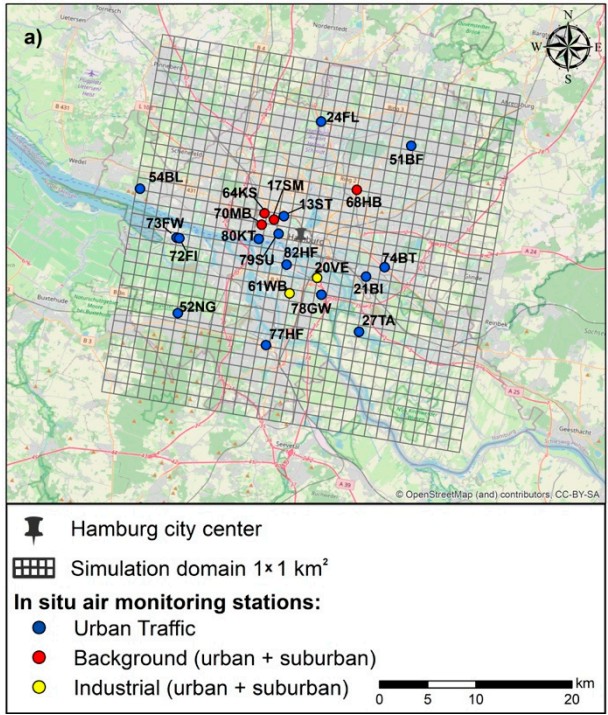 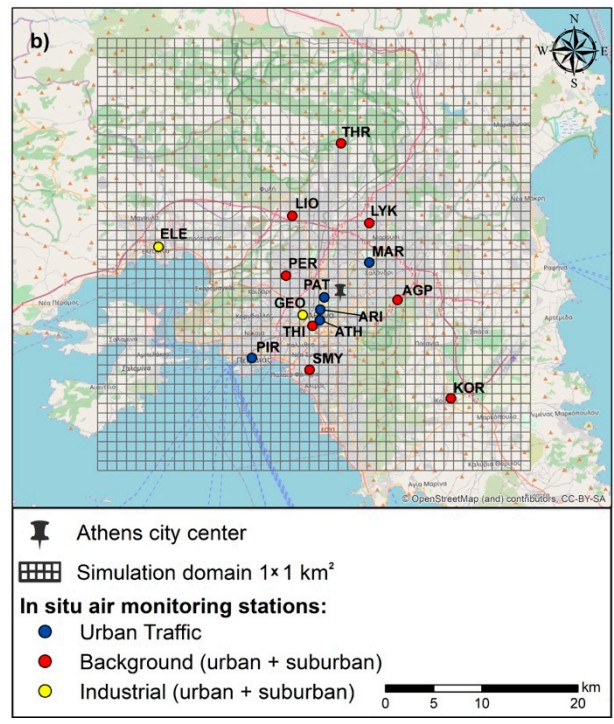

**Figure 5.** The cities of (**a**) Hamburg in Germany and (**b**) Athens in Greece. The gridded rectangles correspond to the air quality simulation domains. Sites correspond to the air quality measurement stations during the simulation periods, which are categorized by type.

### 3.1.1. The Hamburg Demonstrator

The Free and Hanseatic City of Hamburg (Figure 5a) is a federal state and the second-largest city in Germany with a population of over 1.8 million people. Hamburg lies at the River Elbe, which connects the city with the 110 km distant North Sea, making it the gateway to Europe's third largest international port. According to the municipality of Hamburg, road traffic and shipping have been contributing with 35% and 38% to total $NO_X$ emissions as well as 30% and 17% to total $PM_{10}$ emissions in 2012, making the transportation sector the largest emission source for $NO_X$ and $PM_{10}$ [33]. Although overall air quality in Hamburg has improved in recent decades, there are still exceedances of $NO_2$ limit values as defined by air quality standards of the European Union. In Hamburg, there is a well-established air quality monitoring network (Hamburger Luftmessnetz, HaLM, http://www.luft.hamburg.de, accessed: 23 October 2021) continuously measuring air pollutants such as $NO_2$, $NO$, $PM_{10}$, $PM_{2.5}$, $SO_2$, $CO$, $O_3$, and a several heavy metals. It

consists of about 15 measurement stations at traffic, industry, and urban background sites. Since $PM_{10}$ and $NO_2$ limit values were exceeded in recent years, especially at traffic stations, the municipality of Hamburg established its first Clean Air Plan to tackle these exceedances with a variety of actions in 2004 [34]. These plans have been updated in 2012 and 2016.

For Hamburg, there exists a detailed inventory on point sources as derived through mandatory emission reporting under the Federal Emission Control Act, as well as a bottom–up line sources emission inventory which was created and updated for air quality plans in Hamburg 2012 and 2016 [33,35]. The point sources are available every four years (e.g., 2012, 2016), do not cover all criteria pollutants, and are measured, calculated, or estimated. The line sources are based on traffic density and flow data from measurements in combination with traffic-flow modeling and emission factors by HBEFA for the year 2011 [36]. Additionally, there exist scenario calculations for other years. When it comes to other sources such as agriculture, residential heating, airports, etc., emission totals are derived from different sources and gathered in the air quality plan, but no spatial distribution of such emissions is provided. The German Environment Agency offers tools [37] for spatial distribution of nationally reported emissions from different sectors with resolutions of up to 1 km based on different proxies. Nevertheless, such emissions are only available for specific years upon request and are spatially limited to the city's boundaries. Thus, the suitability of such emissions for urban-scale CTMs is very limited.

As a first step to evaluate the UrbEm approach, we compared reported emission totals as given in the Clean Air Plans for 2012 and 2016 with emissions as created with the UrbEm approach from CAMS-REG inventory. Therefore, we applied an urban domain definition for Hamburg in the UrbEm approach, which has an extent of $30 \times 30$ km$^2$ with a resolution of 1 km and covers most of the Hamburg metropolitan area but also some surrounding areas (Figure 5a). In the Hamburg air quality plan for 2016, there are only $NO_X$ emissions reported for the sectors road traffic, industry (incl. energy generation), residential heating, shipping, aviation, railroad, and off-road. When compared to the reported residential heating emissions, the CAMS-REG emissions for the same extent show 25% less $NO_X$ emissions; for shipping, aviation, railroad, and off-road, which can be categorized as the SNAP8 sector, there are 40% less $NO_X$ emissions calculated with UrbEm. In terms of shipping, this high underestimation is mainly due to ship emissions being roughly estimated in the municipality report, while shipping emissions in the CAMS-REG are based on model calculations taking into account AIS-based ship movements [38]. The comparison of road traffic shows slightly higher emissions totals (2%) when applying the UrbEm approach including a factor of 3 to increase road traffic emissions in the urban center. The comparison of reported emissions from industry and energy production with the sum of SNAP sectors 1, 3, and 4 shows 9% higher emissions with the UrbEm approach.

The second step to compare the UrbEm approach with Hamburg as a demonstrator is the analysis of spatial emission distribution. As indicated above, besides detailed information on reported high emitting industrial point sources and emissions from the main road network, there exist no official emissions inventory with spatially distributed emissions.

For the residential combustion sector, CAMS emissions are re-distributed with spatial proxies based on population density. Using these proxies leads to a distribution of emissions that matches the major residential areas in the north of the city center ($PM_{2.5}$ is showed exemplary in Figure 6a,b). In addition, the ratio of relatively high to relatively low emission areas in the population density proxy approach is similar to the respective ratio in CAMS. For the road network sector, CAMS emissions are redistributed with line proxies based on OSM. Thus, the distribution accurately matches the road network of the city of Hamburg (e.g., $NO_X$ displayed in Figure 6c,d). Moreover, first comparisons of road traffic emissions derived with the UrbEm approach to more recent bottom–up modeled emissions based on HBEFA4.1 emission factors [36] and data on traffic density given by the Hamburg municipality revealed good agreement in terms of $NO_X$ and PM emission totals and their spatial distribution, with slight underestimations toward the city center. Spatial proxies for the non-road transport, waste, and agricultural sector are derived from the Corine Land

Cover 2018 dataset [24]. For the non-road transport sector (Figure 6e,f), the high emission areas are now concentrated on the corresponding sources of emissions—Hamburg port and airport Hamburg Fuhlsbüttel. Especially in terms of emissions from port activities by shipping, the spatial distribution is improved compared to the coarser CAMS emissions. Since there are residential areas close to the port areas, this turns out to be a crucial improvement for air quality management assessments. It needs to be mentioned that the Airbus airport is not represented by the CLC proxies, because the Airbus facility is considered an industrial area instead of an airport, although it is frequented by air transport. Redistributed CAMS emissions in the waste sector (Figure 6g,h) are concentrated on high-emission areas in the south of the city center and spatially match waste treatment plants and recycling facilities. Hamburg is surrounded by agriculturally dominated areas. The redistributed emissions in the agriculture sector sufficiently represent the peak emission areas in the southwest of the city ("Altes Land") and in the north and east of the city (Appendix C). In total, the applied spatial proxies in the UrbEm approach to downscale regional CAMS emissions in the Hamburg area match the most important features, which are necessary to represent the spatial distribution of emissions in air quality modeling and management.

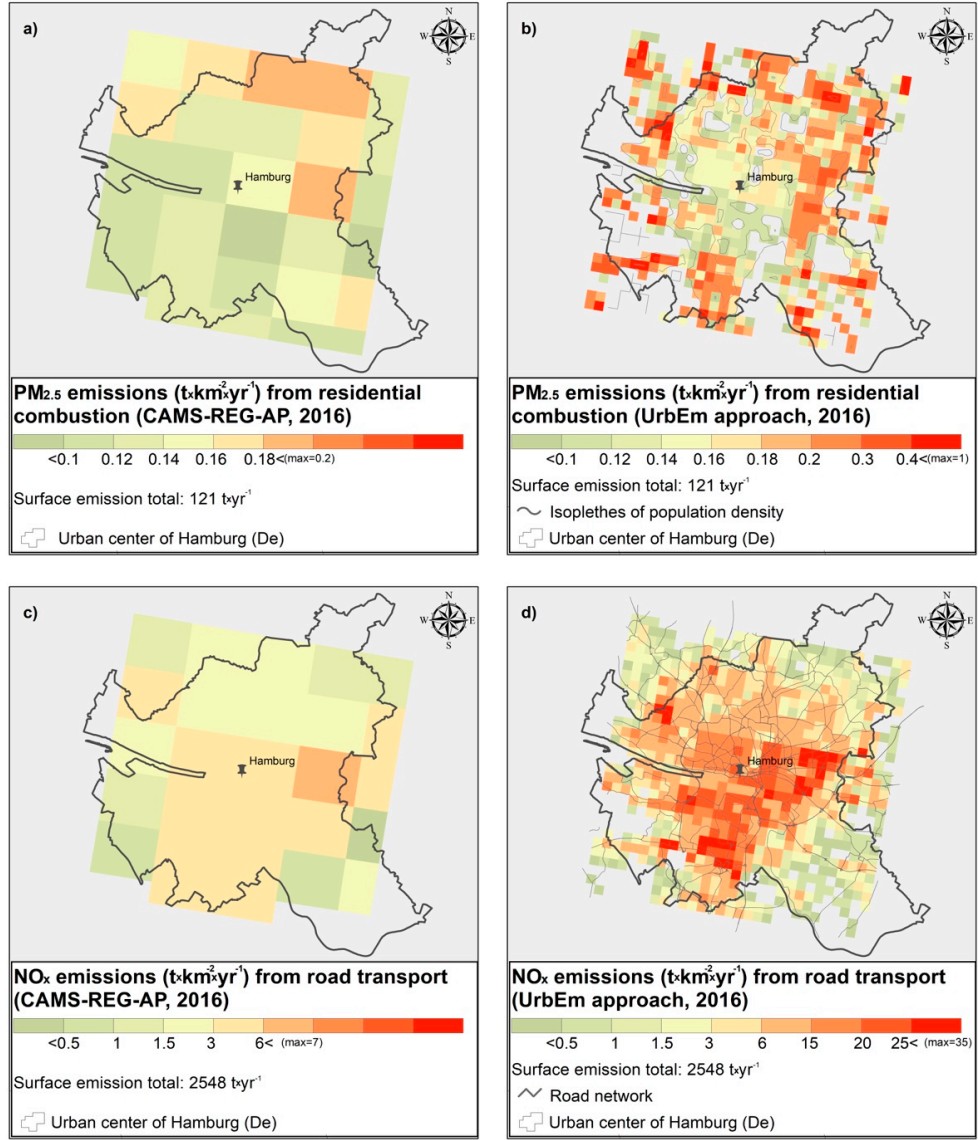

**Figure 6.** *Cont.*

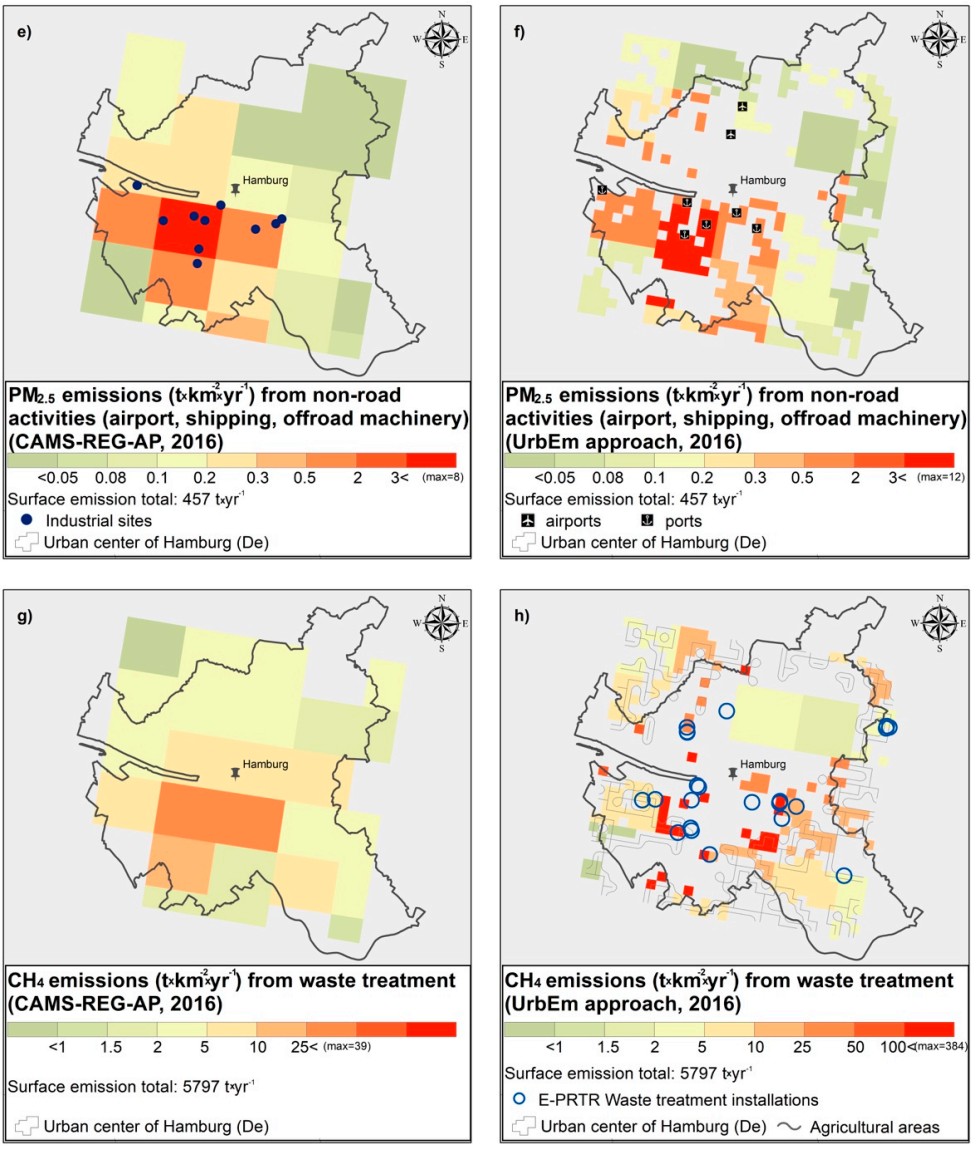

**Figure 6.** Representative air pollution emission fields for Hamburg, as originally provided by CAMS (**a**,**c**,**e**,**g**) and as produced by the UrbEm framework (**b**,**d**,**f**,**h**). Symbols and isopleths of the main proxies used per source type for the spatial disaggregation of CAMS toward the 1 km by 1 km grid are shown on the maps of the right column (UrbEm outputs).

### 3.1.2. The Athens Demonstrator

Athens (Greece) is one of the biggest urban agglomerations in the eastern Mediterranean, which is bound by medium-altitude mountains and by the Saronic Gulf to the south–southwest [39,40]. The Athens basin (450 km$^2$) concentrates the majority of the population (more than 3 million) and economic activity of the Greater Area of Athens, which combined with intense primary emissions, complex topography, and meteorology leads to high levels of atmospheric pollutants [41–43].

A large number (over 2.8 million) of vehicles are circulating in the area, of which approximately 90% are gasoline-powered private cars [42]. Road transport is responsible for a substantial part of not only NO$_X$ but also PM$_{2.5}$ emissions [39,44]. The persistent economic recession has led to a shift in the fuel used for residential heating from fossil fuel toward biofuels, primarily wood, which in turn has caused pronounced wintertime air quality events, especially regarding particle pollutants [41,45]. The major non-traffic, non-heating activity affecting the Athens basin is shipping at and around the port of Piraeus,

which is the largest passenger port in Europe and the second-busiest container port in the Mediterranean. Industrial activity inside the Athens basin is limited [39,42].

Over the last decades, $NO_2$ and PM levels have gradually decreased, although frequent smog events (and violations of the limit values) occur, mostly during wintertime [41,43,46,47]. The air quality of Athens is regularly monitored by the National Monitoring Network (Figure 4b). In situ measurements are also routinely conducted at the Thissio (THI) Air Monitoring Station (37.97326° N, 23.71836° E, 105 m a.s.l, 4 m a.g.l.) of the National Observatory of Athens (NOA), which is considered representative of background conditions in the central Athens basin [48–50].

Athens lacks official gridded and total emission data to compare with the UrbEm result for Athens. To this end, the scientific literature was explored, while applying the algorithm for 2016, for an extent of $45 \times 45$ km$^2$ ($1 \times 1$ km$^2$ cell size) centered at the urban center (official definition by JRC), including both the neighborhood industrial area (at the west) and the national airport (at the southeast). The current comparative findings add value to the evaluation of the spatial allocation of the official national totals at the urban level, which are performed uniformly over Europe by the CAMS methodology [18]. To the best of our knowledge, such comparisons are not yet published for any urban area of the Eastern Mediterranean.

A literature search revealed several national and/or local gridded emission inventories [51–56]. Despite the divergence in the year of reference (from 1990 to 2010), surface coverage (e.g., Attica prefecture, Greater Athens Area, Athens basin, urban compound/area), and pollutants ($NO_X$, $PM_{2.5}$, $PM_{10}$, NMVOC, CO) of these emission datasets, the total CAMS $NO_X$ emitted from the urban center of Athens for 2016 (27,774 t yr$^{-1}$) is consistent with the respective total for 1990 (44,200 t yr$^{-1}$, according to Zachariadis et al. [56]) decreased by the officially reported factor (approximately 0.4) from 1990 to 2016 for Greece reports (EIONET Central Data Repository, https://www.eionet.europa.eu, accessed: 23 October 2021). Similar consistency is also found for $PM_{2.5}$ between 1998 (4888 t yr$^{-1}$, according to Economopoulou and Economopoulos [51] and 2016 (2315 t yr$^{-1}$), using the officially reported decrease factor of 0.5. The industrial sector CAMS $NO_X$ emission totals for Athens (21,139 t yr$^{-1}$) are similar to those reported by Markakis et al. [52] for 2003 (22,400 t yr$^{-1}$). For road transport emissions, the comparison among datasets is more challenging. CAMS urban center totals (using the EIONET conversion factors from 1990 to 2016) are significantly lower than all published totals emitted from Athens. The modeling analysis performed by Kuik et al., 2018 [28], suggests that traffic emissions by CAMS are strongly underestimated, which is in alignment with the aforementioned discrepancy and is partly attributed to the inadequate representation of traffic congestion. A distinct difference among the spatial distribution of emission totals stems from the expanded urban area for 2016 when compared to all previously published maps.

Focusing on the performance of the spatial disaggregation approach, we have developed comparative maps of the regional CAMS, and the local scale UrbEm emissions from Athens sources are given in Figure 7. Some findings apply to all types of sources and are due to the complex topography of Athens. For example, residential combustion emissions from coarse resolution grid cells (Figure 7a) extend beyond the coastline, whereas the use of population density as a high-resolution proxy to redistribute this sector eliminates this error (Figure 7b). Along the same line, the algorithm attributes all emissions from anthropogenic activities around mountainous areas at the downhill urbanized cells (e.g., road transport; Figure 7c,d). As evident through the mass totals (given at the bottom of each plot in Figure 7) and the spatial distribution of emissions within each coarse cell, although mass is conserved, the range of emission values is amplified by a factor of ca. 5 (road transport and waste treatment) or 10 (residential heating and non-road activities) for the high-resolution mapping. The pronounced gradient of UrbEm spatial fields is definitely an improvement stemming from the appropriate and high-resolution spatial proxies, which are used to allocate CAMS gridded emissions. The proxies are contemporary and frequently updated

to ensure the consistency with the actual urban landscape during the period of study. The efficiency of this optimization is quantified in the next section.

Examining each source sector separately, the UrbEm approach depicts spatial features of the air pollution emissions from residential combustion sources that coarse mapping fails to capture (Figure 7a,b). In particular, emissions are allocated at the inhabited areas in and around the urban center, with maximum values in the northern suburbs and in the mountainous residential areas. Most of the particulate matter emitted from residential combustion activities is known to originate from biomass (wood) burning, according to CAMS data for Greece. The latter (in particular TNO-MACC II [57]) has been proven reasonable for Athens during the economic crisis [48]. Therefore, the specific high-resolution dataset is expected to accurately represent residential wood burning emissions in Athens from 2009 to the present.

The biggest asset of the developed tool lies beyond the hybrid mechanism to produce the spatially disaggregated road transport emissions, either in fine ($1 \times 1$ km$^2$) cells or attributed to the contemporary road network of the studied city. As expected, the spatial allocation of vehicle emissions through CAMS (Figure 7c) is performed with consistency to the density of the Athens road network (Figure 7d). The finer disaggregation applied through UrbEm reveals spatial gradients and emission maxima from roads in more detail and accuracy. The optimization of air pollution from road transport is realized when the emitted masses are ascribed to lines and then combined with air quality models that incorporate urban canyon processes (see Section 3.2).

Waste treatment emissions are known as more localized than how they are represented by CAMS (Figure 6e). With the usage of the locations of waste treatment installations, mass allocations are performed in higher accuracy (Figure 7f). The open burning of waste has an increased positioning uncertainty, as it can potentially occur on any agricultural surface.

CAMS data representing emissions from non-road activities (Figure 7g) are greatly improved by allocating airport and shipping emissions at their actual land and sea surfaces (Figure 7h). Shipping is indeed found as a pronounced source of air pollution, which is consistent with what is stated above for Athens. With respect to non-road machinery, its allocation follows the same restrictions mentioned for open waste fires.

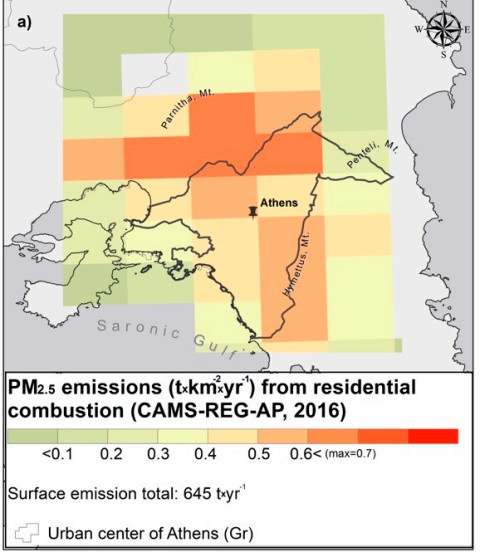 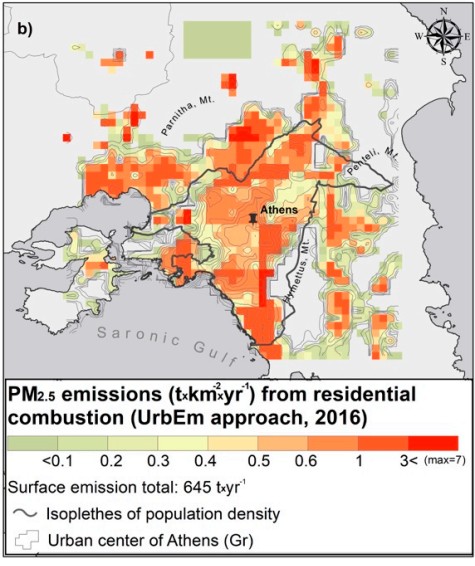

**Figure 7.** *Cont.*

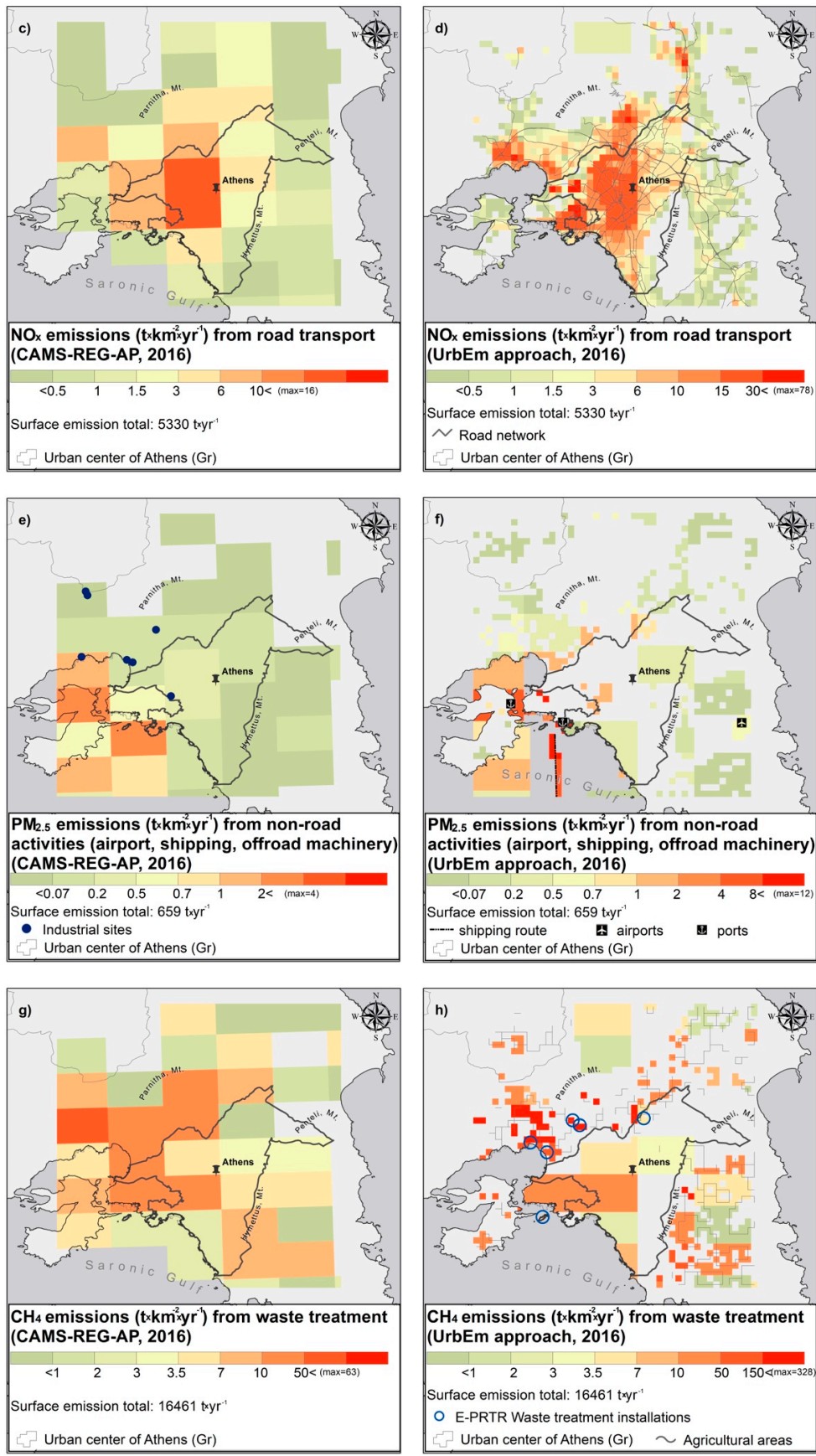

**Figure 7.** As in Figure 6, but for Athens.

## 3.2. Verification through Air Pollution Predictions

The evaluation of the developed approach (UrbEm) to arrive at urban, high-resolution emission inventories is performed through their implementation in atmospheric numerical simulations. In particular, AQ predictions for Hamburg are performed for 2016 and compared with observations from 13 and five stations measuring $NO_2$ and $PM_{2.5}$, respectively (Figure 4a). For Athens, a heavily polluted month (December 2018) is used to demonstrate the value of high resolution (including linear road) emissions for a city that faces complex challenges, as described in Section 3.1.2. $NO_2$ and $PM_{2.5}$ observations made by the National Monitoring Network (Figure 4b) and at the supersite at Thissio [50] are compared with the respective predictions. In both demonstrators, the urban stations are categorized and grouped as traffic, background, and industrial.

In order to quantify the assets of the UrbEm approach discussed in Section 3.1, namely the use of high-resolution spatial proxies for area emission distribution, plus the allocation of road transport emissions in line sources, the CAMS (no proxy) and UrbEm emission datasets already discussed in Section 3.1 stand as the core of the two emission scenarios applied in both demonstrator cities (CAMS no proxy and UrbEm scenario, respectively). Based on the already mentioned findings of Kuik et al. (2018) [28], the proposed multiplication factor of 3 is applied in the emissions by road transport. As this was found to remediate underestimations in regional emission inventories (as well as other possible sources of model biases), it is also applied in the CAMS no-proxy scenario. Such a consistency keeps comparisons among scenarios linked to their spatial divergence.

Hourly emission rates are calculated by using the monthly, weekly, and diurnal profiles indicated through Bieser et al. (2011) [58] for each source sector. The monthly profiles for road transport in Athens are adjusted according to national data [59]. Daily and hourly profiles for residential heating are based on air pollution measurements in Athens [48]. Additionally, area emissions of residential heating are generated with dependence on the daily average temperature in Hamburg and Athens. Lastly, the split of $NO_X$ (CAMS output) to NO and $NO_2$ emissions (EPISODE-CityChem inputs) is 95:5 for area and point and 70:30 for line emissions.

Emission databases correspond to reference years. In order to provide high-resolution emissions for any year of interest, the official, national reports (EIONET Central Data Repository, https://www.eionet.europa.eu) of total emissions are compared. The extracted multiplication factors per source and pollutant are applied and convert emissions from the year of reference to the year of interest.

### 3.2.1. Chemistry Transport Model Setup

EPISODE-CityChem is a Chemistry Transport Model to enable chemistry/transport simulations of reactive pollutants on the city scale. In particular, the horizontal spatial resolution of simulations is 1 km with an embedded regular receptor grid (100 m × 100 m), which hosts Gaussian line and point source dispersion and local photochemistry [22]. More details on the features of the model and its configuration and setup for the current study can be found in Appendix B.

### 3.2.2. Comparison of Predictions and Observations

The comparison with available observations (sites of which are shown on Figure 4) support the efficiency of the UrbEm approach, especially for $NO_X$. In particular, the EPISODE-CityChem model is found to underestimate observations (Appendix C) in both cities, but the intercomparison of the average mean biases from CAMS no proxy to UrbEm shows a significant reduction of this underestimation by more than 100% in most cases. Indeed, $NO_2$ predictions from these two model runs have been compared, and their difference is found statistically significant at the 95% confidence level ($p < 0.05$) for all stations in both cities. The temporal correlation ($r$) among measurements and observations is significantly improved when replacing the CAMS no proxy and the UrbEm approach. As expected for urban environments, the deviation of observations from their mean value

is elevated, which cannot be represented when emissions are coarsely allocated to their sources (CAMS no-proxy scenario). In contrast, UrbEm enables the depiction of a greater dispersion of air pollution around the mean urban values, which is reflected through the higher standard deviation found, better matching the real case in both cities. In addition, the fraction of modeled values within a factor of two of the observed values is in most stations above 50%, reaching 75% for $PM_{2.5}$, when the UrbEm scenario is considered. The greatest improvement of model performance occurs for $NO_2$ over the industrial and traffic urban areas of Athens as well as for Hamburg (Figure 8). This is attributed to the strong spatial variation of urban land types, and thus of emission sources, when combined with elevated anthropogenic emissions during wintertime, the short lifetime, and strong gradients of $NO_2$. This combination necessitates the right spatial allocation of urban emissions, namely at the road network and the industrial sites generating much stronger gradients when compared with a coarser resolution grid.

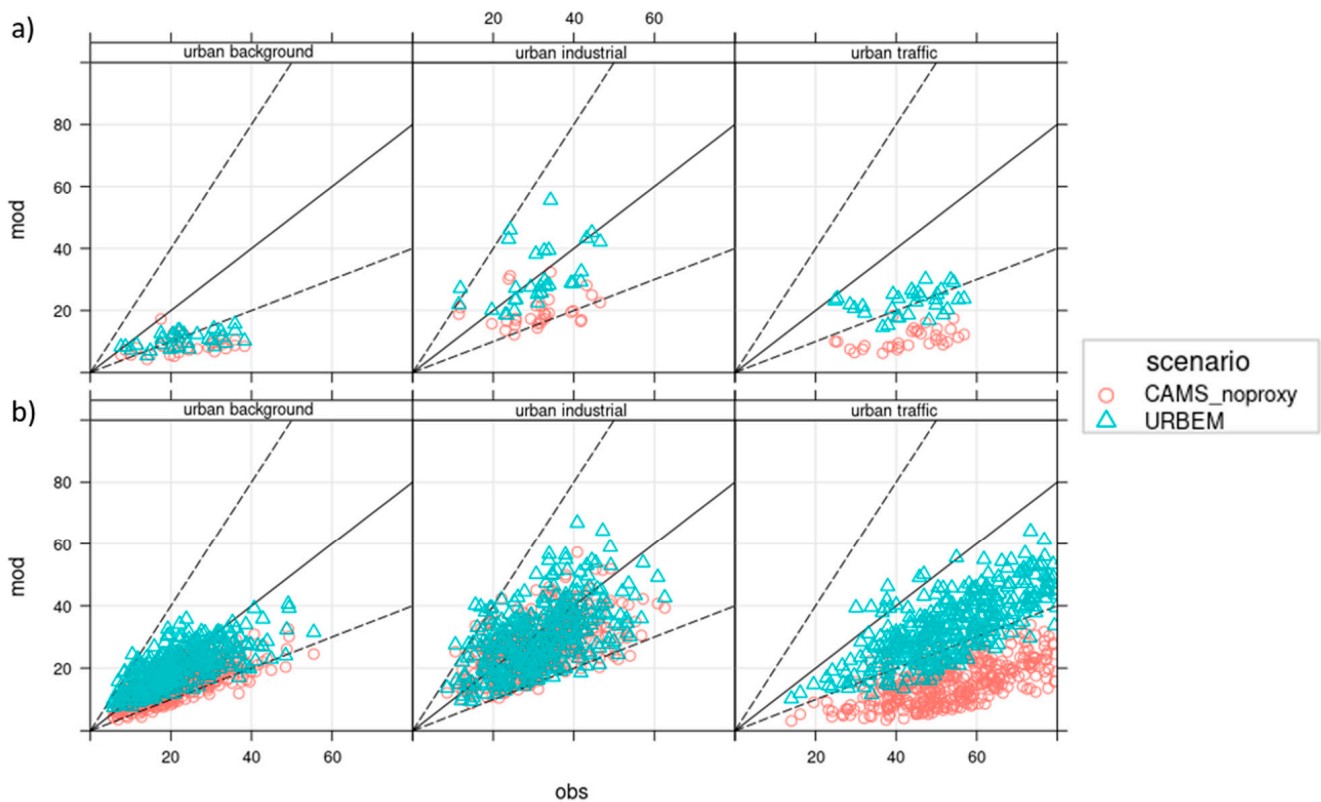

**Figure 8.** Daily $NO_2$ (µg m$^{-3}$) scatter plots for (**a**) Athens, December 2018 and (**b**) Hamburg, 2016. The observed (obs) versus modeled (mod) mass concentrations are shown, the latter when applying the original CAMS emissions (CAMS no proxy) and high-resolution emissions using the newly developed approach (UrbEm).

In order to proceed to a targeted inter-comparison between the two model scenarios/disaggregation approaches, Figure 8 (and Appendix C) presents the scatter plots to show the deviation of model predictions for mean daily $NO_2$ (and $PM_{2.5}$) from the observed values (markers upon the solid 1:1 line) up to a factor of two (markers inside the dashed 1:2, 2:1 lines) and further (markers outside the dashed lines). An overall statement is the comparative advantage of the UrbEm approach, shifting the cloud of observation–prediction pairs toward the 1:1 line, which is striking for $NO_2$. This performance is more apparent in emission hotspots (traffic or industrial areas) and is explained by their more accurate spatial representation by the UrbEm approach. Especially for the urban traffic sites, the significance of the line representation of road transport emissions is revealed through the distinct position of the CAMS no proxy outside the dashed lines. Through high-resolution emission allocation, mass rates from their actual hotspots (e.g., roads, industrial sites, resi-

dences) are increased; thus, concentration underestimations over and downwind primary emissions are decreased.

The effects of the UrbEm approach on $PM_{2.5}$ model performance are less pronounced (Appendix C). This is expected over and downwind urban traffic areas, because traffic has a smaller footprint on $PM_{2.5}$ and hence on its spatial disaggregation. More broadly and independent of pollution sources, $PM_{2.5}$ shows less spatial variability due to its governing atmospheric processes, including the role of secondary inorganic and organic formation of atmospheric aerosols. Overall, the applied downscaling is especially valuable for short-lived species, but in some cases, improvements can be seen also for $PM_{2.5}$. It is worth mentioning that these improvements are important, given the statistical comparison performed among the two model scenarios, which found their difference significant at the 95% confidence level ($p < 0.05$) for all stations in both cities.

Focusing on the air pollution mapping over the complex area of Athens during a highly polluted winter month, it seems that the spatial optimization of coarse emissions contributes more than 30% and up to 70% to the modeled $NO_2$ concentrations over most of the urban area (Figure 9a) and the other polluted areas outside (e.g., the industrial area in the west and the national road toward the north). Indicatively, over the sites of model-measurement mean monthly inter-comparison (e.g., PIR, ELE, PER, LIO), the UrbEm approach improves model results by 30–50% when compared to the CAMS no-proxy scenario.

The contribution of UrbEm to the mean monthly $PM_{2.5}$ predictions (Figure 9b) based on CAMS emissions is lower (up to 30%) and occurs mainly outside the urban center, over the industrial areas and the northern residential suburbs. These maxima of UrbEm effects are related to the spatial optimization of industrial emissions and residential combustion, respectively. From the four measuring sites of $PM_{2.5}$, only the predictions at the urban traffic site at the port (PIR) seem to differentiate around 20%, with the UrbEm approach being consistent with the mean monthly observations.

Figure 9c,d illustrate the differences between the CAMS no proxy and UrbEm for both $NO_2$ and $PM_{2.5}$ concentrations in Hamburg. Especially for $NO_2$, the linear structures of the road network become more distinct in the UrbEm approach. In general, the UrbEm approach allows a wider distribution of concentrations in the city domain. For both $NO_2$ and $PM_{2.5}$, it is expectable that the concentrations peaks are in the city center and the port area, which is additionally a hotspot for industrial activities. For further discussion of air pollutant concentration for Hamburg in 2016 as derived from emissions downscaled with an early version of the UrbEm, we refer to Ramacher and Karl (2020) [29].

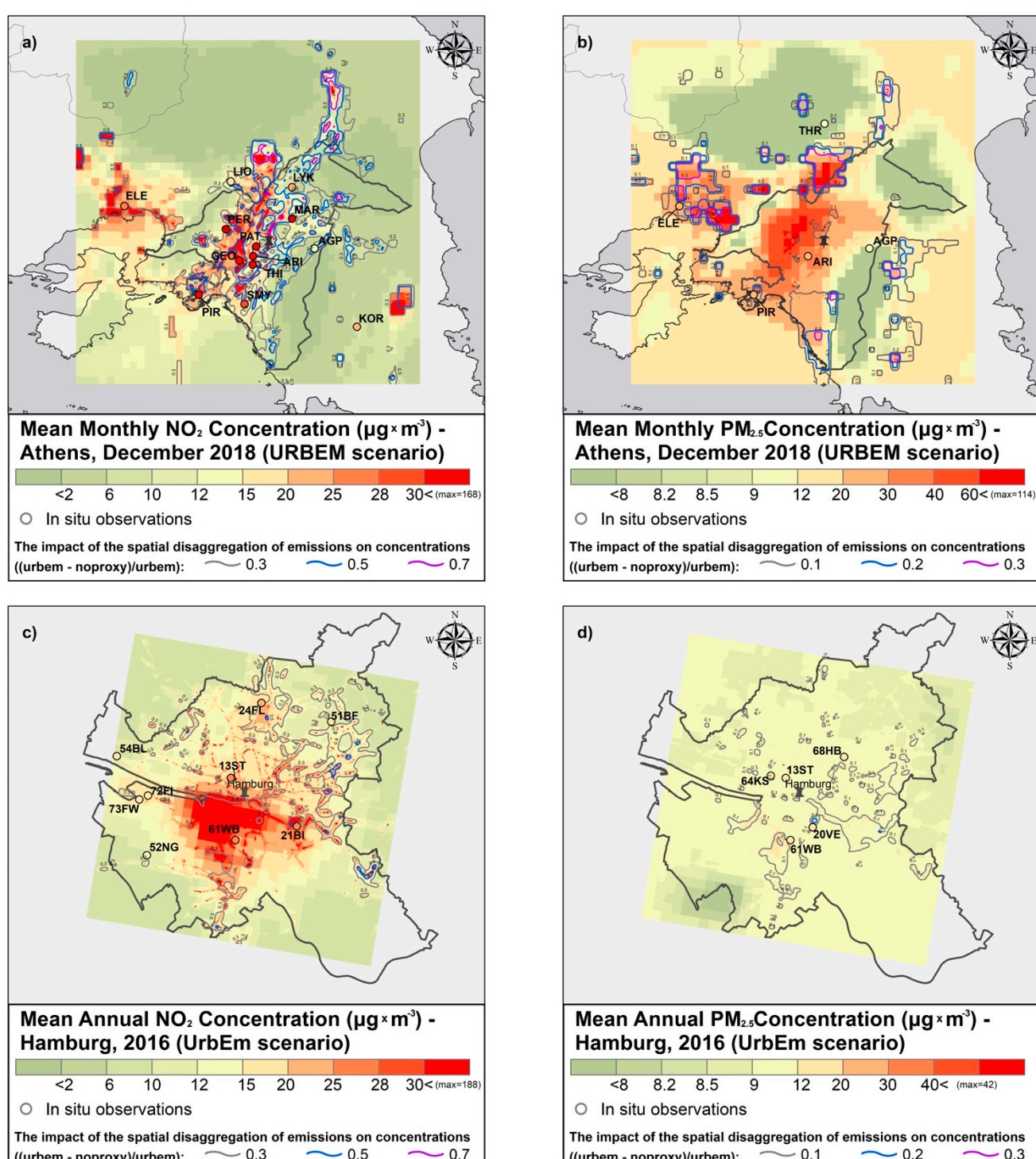

**Figure 9.** Map plots of CTM AQ ($\mu$g m$^{-3}$) results for (**a**) NO$_2$, (**b**) PM$_{2.5}$ in Athens, and (**c**) NO$_2$, (**d**) PM$_{2.5}$ in Hamburg. The difference (0–1) of concentrations between the scenarios is shown in isopleths. The colored circles are the measured values.

## 4. Discussion

The comparison of downscaled regional CAMS-REG emissions ($\approx$6 $\times$ 6 km$^2$ grid cells) with the introduced UrbEm approach to the same emissions that were mapped to our high-resolution grid without the use of any spatial redistribution proxies showed substantial changes in the fields of emissions in Hamburg and Athens. The city-specific exploration of local features, such as waste and recycling facilities, road networks, industrial and

agricultural areas, as well as the locations of ports and shipping lanes guided and optimized the spatial distribution of connected emissions through the use of the appropriate, detailed spatial proxies in the frame of the UrbEm approach. The approach is of utmost importance for modeling air quality for coastal cities, as it ensures zero emissions from the sea surfaces that are adjacent to land-oriented emissions and vice versa. Despite the claim of the UrbEm approach to be generally applicable in European cities, there are some remaining imprecisions in the spatial distribution, e.g., the consideration of all types of airports, independent of their use (e.g., public or military) or the specific location of point sources with high emissions that are not covered by the E-PRTR emission inventory because they are below the reporting thresholds. Thus, knowledge, if possible, on the local emission features, can optimize the outputs of the UrbEm approach, as well as reveal desired deviations from the horizontal CAMS methodology at the domains/cities under study.

Considering the treatment of road traffic emissions, the UrbEm approach allows for distributions of emissions to any city's road network by using OSM data. Evidently, that improves the distribution of road traffic emissions when compared to grid emissions. Thus, this feature is of high value for any detailed spatial analysis of urban air pollution, while it can be essential for urban-scale air quality models capable of handling road emissions from line instead of area sources. The coupling of UrbEm with Episode-CityChem for two city demonstrators showed an increased accuracy and a better representation of gradients in intra-urban air pollution predictions. According to a recent sensitivity study on the choice of the subgrid resolution in urban-scale air pollution modeling [60], the 100 m resolution (used here) has been found competent to represent near-road concentration estimates. Nevertheless, model underestimations at road traffic sites remain both in Hamburg and Athens, even though we applied the emission correction factor of 3 proposed by Kuik et al. (2018) [28] to all pollutants in both cities. A possible reason might be the ventilation effects and concentration agglomerations in street canyons or at junctions that might not be adequately represented in chemistry transport modeling. Another source of discrepancy could be the uniform application of the NO to $NO_2$ ratio, which was applied to split the $NO_X$ input provided by CAMS. Additionally, occasional heavy traffic at intersections and traffic jams is not taken into account when conventional temporal emission profiles are applied. Similarly for the spatial dimension, the distribution of regional road traffic emissions to urban-scale road links as derived from OSM does not take into account city-specific traffic densities, congestion, or other spatial features that may be applied in bottom–up approaches. Lastly, a frequent source of model discrepancies is the non-accounted air pollution sources in emission inventories, such as the recirculation/resuspension of particulate matter due to vehicle movement [48,53].

The UrbEm approach does not claim to replace city-specific emission modeling efforts or bottom–up emission inventories. Nevertheless, the creation of bottom–up anthropogenic emissions is still a demanding and expensive task due to the necessary collection of city-specific data (e.g., road-specific vehicle densities, fleet compositions) and their compilation toward emission rates per source and pollutant. In addition, most of the detailed city-specific emission inventories are available for specific years and/or upon request; these are spatially aggregated or their disaggregation is frequently in cells above 2 km spatial resolution, road (and occasionally industrial) emissions are attributed to cells (instead of lines or points), and it is limited to municipal boundaries or prescribed domains. In cases, specific sources and/or pollutants could be missing from existing local inventories. In addition, for big cities, such as Athens and Hamburg, there are lots of high-resolution data for adequate urban air quality modeling temporally and spatially aggregated or missing. Lastly, the scope of city-scale inventories may be different from typical national or European-wide inventories, which makes it difficult to nest the city-scale inventory into a wider regional inventory [61].

Therefore, the UrbEm approach enables urban air quality planners and modelers that lack resources and city-specific information to initiate efforts in air quality management. For cities such as Athens, with a complex terrain and official emissions at the national level,

the UrbEm approach to downscale open regional emissions adds value to air quality management efforts by using the detailed spatial proxies in downscaling. Taking into account that the spatial data that underlie the applied proxies in the UrbEm approach are frequently updated and/or become more detailed, we expect the spatial representativeness of emissions downscaled with UrbEm to enhance. Indicatively, once the building height data by CLMS (https://land.copernicus.eu/local/urban-atlas/building-height-2012, accessed: 23 October 2021) become available for all European cities (rather than solely capitals), its incorporation to our approach will certainly optimize the spatial allocation of residential heating emissions.

Through this work, it is shown that the high-resolution emission allocation with UrbEm improved the performance of modeled values from a state-of-the-art city-scale CTM. Such an approach provided also flexibility in the selection of the region, resolution, and period of interest for AQ modeling. Lastly, comparative modeling studies among several European cities is feasible and concrete when using gridded emission data produced by the same methodology and tool, based on a common European database, thus aligning with the call from the European Commission's recent Air Quality—Fitness Check of the AAQ Directives for further harmonizing modeling activities across the EU [62].

Overall, the investment upon existing knowledge and data hosted on reliable programs/platforms such as Copernicus/CAMS is an approach to overcome the lack of bottom–up high-resolution emission inventories for certain cities, or it can be an equivalent alternative to pre-existing local emission data. Whichever the case, such exploitation and development can certainly act as feedback toward the continuous improvement and optimization of the provided services.

## 5. Conclusions

The use of regional emission inventories can be challenging for urban-scale AQ applications and air quality management in cities. Nevertheless, their exploitation through disaggregation by utilizing spatial proxies, available on a European or global scale, is a credible solution for European cities that lack bottom–up emission inventories. To this end, we developed the UrbEm approach, which enables in a modular manner the downscaling of gridded regional emissions with specific spatial proxies based on a variety of open access, robust, sustainable, and frequently updated sources. The presented approach can be applied to any urban area in Europe and provides methodological homogeneity between different cities.

Prioritizing applicability and homogeneity, the developed approach avoids any dependence on occasional, local, high-resolution data, without excluding this option once such data (e.g., variable traffic volumes for specific road segments) are available for cities or periods of interest. Nevertheless, the method invests upon well-established European emission inventories (here CAMS-REG), which already have incorporated pollutant- and source-specific data (e.g., traffic densities, vehicle and road types), for the spatial disaggregation of national emission totals. All data that support the high-resolution spatial proxies used are being quality controlled and assured (see more on Appendix A), while the limitations relevant to CAMS-REG or any traditional top–down emission inventory are expected unless near real-time emission data were ideally available and used.

To demonstrate the general applicability and performance of the developed method and tool, we compared the spatial distribution of uniformly disaggregated regional emissions with emissions downscaled with the UrbEm approach for the differing cities of Athens and Hamburg. The representation of local features in general and of emission hotspots in particular was largely improved. We further applied the two emission inventories in a state-of-the-art urban-scale Chemistry Transport Model and compared the results against measurements. Urban-scale air pollutant concentrations based on the UrbEm approach show improved agreement, especially for $NO_2$ and measurement sites close to road traffic.

The UrbEm downscaling approach is completely free of cost and open source, which is accompanied with an efficient, fully automated, and intuitive tool (https://github.com/martinottopaul/UrbEm, accessed: 23 October 2021) that enables users to create high-resolution urban emissions from point, line, and area sources. We are confident that UrbEm fully covers the emission inventory prerequisite for all urban air quality modeling and management efforts and serves as a starting point for such efforts in cities across Europe.

While the work described here establishes UrbEm, we envision a continuous improvement of the methodology regarding its technical components such as the integration of improved proxies. Beyond the paradigm of the building height data by CLMS discussed already, the Automatic Identification System (AIS) data used for shipping emissions is such a candidate. The latter was exploited under the uEMEP implementation [60], which is a similar undertaking to ours and with whom we will pursue technical exchange regarding the use of proxies not currently handled by them. Moreover, comparisons between the two methodologies and with other bottom–up implementations in European cities would deliver a clear insight on the balance between overall resources required and efficiency across Europe. Relevant to this, comparative applications of atmospheric numerical models using UrbEm and bottom–up local inventories when and where available would further build the proof of efficiency of the method as well as reveal deficiencies to alleviate. Lastly, we plan to expand the automization of the whole workflow, minimize the need for location-specific refinements, and keep the process per city temporally efficient. A valuable addition would be the integration of the newly developed temporal profiles CAMS-TEMPO [63] toward producing hourly, instead of yearly, emission rates, based on spatially-variant temporal profiles.

On a more policy-related note, UrbEm aims at contributing to the current discussion on the Revision of Ambient Air Quality Directives [64] (https://ec.europa.eu/environment/air/quality/revision_of_the_aaq_directives.htm, accessed: 23 October 2021) and its ensuing policy implications. In particular, we aspire to support—Europe-wide—the mitigation of administrative burden of air quality management by offering such an agile tool, to increase the comprehensiveness of air quality assessments by providing a common staring point, and to bring out the underlying aspects of environmental inequalities present in the spatial distribution of emissions, concentrations, and exposure in urban locales while offering a credible solution for planning equitable measures.

**Author Contributions:** Conceptualization, M.O.P.R., A.K., O.S. and E.A.; Data curation, M.O.P.R. and A.K.; Formal analysis, M.O.P.R., A.K., O.S., M.K., R.T., H.D.v.d.G., J.K., E.G. and E.A.; Funding acquisition, E.G. and E.A.; Investigation, M.O.P.R., A.K., O.S., J.F., M.K., R.T., H.D.v.d.G., J.K., E.G. and E.A.; Methodology, M.O.P.R., A.K. and E.A.; Project administration, E.A.; Software, M.O.P.R. and A.K.; Supervision, M.O.P.R. and E.A.; Validation, M.O.P.R., A.K., J.F. and E.A.; Visualization, M.O.P.R., A.K. and J.F.; Writing—original draft, M.O.P.R., A.K., O.S., J.F. and E.A.; Writing—review and editing, M.O.P.R., A.K., O.S., J.F., M.K., R.T., H.D.v.d.G., J.K., E.G. and E.A. All authors have read and agreed to the published version of the manuscript.

**Funding:** The authors acknowledge the funding received by ERA-PLANET (www.era-planet.eu, accessed: 23 October 2021), trans-national project SMURBS (www.smurbs.eu, accessed: 23 October 2021) (Grant Agreement n. 689443), funded under the EU Horizon 2020 Framework Programme.

**Data Availability Statement:** Two different software solutions, written in R and Python respectively, for the UrbEm approach are available for download at https://github.com/martinottopaul/UrbEm (accessed: 23 October 2021), including necessary input data as well as guidance to download necessary databases. The produced emission inventories and results of air quality simulations for Athens and Hamburg are available upon request.

**Acknowledgments:** The computational time spent for the AQ model applications in Athens was provided by the Greek Research and Technology Network (GRNET) in the National HPC facility—ARIS—under project ID CiTyM. N.K. is grateful for the systematic collaboration with Themistoklis Kontos from the department of Environment of the University of the Aegean (Greece), who provided support and advice for the GIS mapping.

**Conflicts of Interest:** The authors declare no conflict of interest.

## Appendix A. Short Description of Spatial Datasets

### Appendix A.1. European Pollutant and Transfer Register (E-PRTR)

E-PRTR (http://prtr.ec.europa.eu/, accessed: 23 October 2021) is the Europe-wide register that provides key environmental data for industrial facilities in European Union Member States and a few other countries, and it provides annual emission totals for large sources i.e., for pollutants that exceed limit values. In addition to the annual emission totals, the geographic location and sector for these sources is also defined. The official national data transmitted by the Member States to the European Environmental Agency (EEA) are subjected to a quality control through an automated validation tool developed by the EEA (https://eur-lex.europa.eu/legal-content/EN/TXT/PDF/?uri=CELEX:52017SC0710&from=EN, accessed: 23 October 2021).

The emissions of nitrogen oxides ($NO_X$), sulfur oxides ($SO_x$), carbon dioxide ($CO_2$), ammonia ($NH_3$), and particulate matter ($PM_{10}$) from industrial facilities are utilized in this work. In addition, geo-locations from this database are selectively used for the spatial disaggregation of emissions from relevant sources (e.g., public power; see Table 1).

### Appendix A.2. The CAMS Regional Anthropogenic Emissions (CAMS-REG)

CAMS provides sectoral annual emission totals for Europe (30° W–60° E and 30° N–72° N) for 2000–2015 (CAMS-REG-v2.2.1 or CAMS-REG-v1 for 2003–2009) and 2016 (CAMS-REG-v3.1). In particular, these databases involve annual emission rates of $CH_4$, CO, NH3, NMVOC, $NO_X$, $PM_{10}$, $PM_{2.5}$, and $SO_2$ from road, air, rail transport, navigation, mobile machinery, fuel production, industrial activities (paper, cement, minerals, metals, etc.), stationary combustion, agriculture, waste, solvent use, and public power. The first two versions are based on the SNAP sector classification and provide emission rates at the horizontal spatial resolution 0.125 × 0.0625 degrees (lon/lat) [17,57], while CAMS-REG v3.1 has adopted the GNFR sector classification and provides data at 0.1 × 0.05 degrees [18].

In the core of CAMS methodology lies the national reports of emissions by year, pollutant, and sector registered in and available through the Centre for Emission Inventories and Projections (http://www.ceip.at/, accessed: 23 October 2021). Supplementary sources include the Environmental European Agency (EEA), the Greenhouse gas—Air pollution Interactions and Synergies model (GAINS), the Emissions Database for Global Atmospheric Research (EDGAR), and estimates by the Netherlands Organization for Applied Scientific Research (TNO). The spatial distribution of country total emissions from point sources is based on the E-PRTR database, while non-point sources are disaggregated through spatial proxies, such as total, rural, and urban population, arable land, and the TRANSTOOLS road network.

A thorough report on the quality of the spatial data of the CAMS-REG inventory lies in Kuenen et al. (2021). Overall, uncertainties with respect not only in the National registrations but also in the spatial disaggregation proxies and methods used are present and difficult to quantify. With respect to the urban environment, indicative sources of errors include the following: (i) the spatial disaggregation of vehicle emissions is performed for a single year (instead of annually), excluding actual real-time road intensities and small road traffic volumes; (ii) power plant and industrial emission registrations include errors and inconsistencies; (iii) building characteristics (e.g., type of building, level of insulation) are not taken into account for the spatial disaggregation of residential heating emissions; (iv) occasional large construction sites are not taken into account for the construction of off-road machinery emissions.

Focusing on the cities/demonstrators of the present intra-urban disaggregation, the emissions of all pollutants in Hamburg (Germany) is based on CEIP, except for $CH_4$, which is retrieved by EEA. The latter also holds for Athens (Greece), whereas the rest of the gaseous pollutants are based on EEA data. $PM_{2.5}$ and $PM_{10}$ are GAINS estimations.

*Appendix A.3. Corine Land Cover*

The CORINE Land Cover (CLC) inventory (https://land.copernicus.eu/pan-european/corine-land-cover, accessed: 23 October 2021) was initiated in 1985 (reference year 1990). Updates have been produced in 2000, 2006, 2012, and 2018. It consists of an inventory of land cover in 44 classes. CLC uses a Minimum Mapping Unit (MMU) of 25 hectares (ha) for areal phenomena and a minimum width of 100 m for linear phenomena. The Eionet network National Reference Centres Land Cover (NRC/LC) is producing the national CLC databases, which are coordinated and integrated by EEA. CLC is produced by the majority of countries by visual interpretation of high-resolution satellite imagery. The Sentinel 2 images used provide homogeneous, high-quality multi-temporal imagery, to support high-quality identification of land cover changes in Europe (https://land.copernicus.eu/user-corner/technical-library/clc2018technicalguidelines_final.pdf, accessed: 23 October 2021). In a few countries, semi-automatic solutions are applied, using national in situ data, satellite image processing, GIS integration, and generalization.

CLC has a wide variety of applications, underpinning various community policies in the domains of environment, but also agriculture, transport, spatial planning, etc. In this study, we applied CLC classes from the 2018 version as spatial proxies to distribute emissions to different sectors.

*Appendix A.4. Global Human Settlement Layer*

The Global Human Settlement Layer project of the European Commission's Joint Research Centre [23] addresses these needs with spatially detailed information on population and settlements. The GHSL are offered as open and free data. The data range goes from global coverage datasets to pan-European built-up layers (the European Settlement Map) to analytical data (e.g., the Urban Centre Database [30]). In this study, we apply the Global Human Settlement population grid (GHS-POP) and the Global Human Settlement Layer Urban Centres Database (GHS-UCDB).

The GHS-POP spatial raster dataset depicts the distribution of population, expressed as the number of people per cell (https://ghsl.jrc.ec.europa.eu/ghs_pop2019.php, accessed: 23 October 2021). Residential population estimates for target years 1975, 1990, 2000, and 2015 provided by CIESIN GPWv4.10 were disaggregated from census or administrative units to grid cells, which was informed by the distribution and density of built-up areas as mapped in the GHSL global layer per corresponding epoch. The available resolutions are 250 m, 1 km, 9 arcsec, and 30 arcsec. In this study, we apply the GHS-POP layer for 2015 with a resolution of 30 arcsec (WGS84) (GHS_POP_E2015_GLOBE_R2019A_4326_30ss_V1_0) to spatially distribute emissions that are connected to population activity, e.g., residential heating.

The GHS-UCDB is the most complete database on cities to date, which was publicly released as an open and free dataset. The database represents the global status on Urban Centers in 2015 by offering cities' location, their extent (surface, shape), and describing each city with a set of geographical, socio-economic, and environmental attributes, many of them going back 25 or even 40 years in time. Urban centers are defined in a consistent way across geographical locations and over time, applying the "Global Definition of Cities and Settlements" developed by the European Union to the Global Human Settlement Layer Built-Up (GHS-BUILT) areas and Population (GHS-POP) grids. A validation of the global human settlement through the Landsat imagery, as well as a quality control to ensure that all input population was disaggregated and totals were preserved, have been successfully conducted [65]. In this study, we apply the GHS-UCDB layer (GHS_STAT_UCDB2015MT_GLOBE_R2019A_V1_0) to identify the geographic extent of urban centers.

*Appendix A.5. OpenStreetMap*

OpenStreetMap (OSM) [25] is a collaborative project to create a free editable map of the world. The geodata underlying the map is considered the primary output of the project.

Created by Steve Coast in the UK in 2004, it was inspired by the success of Wikipedia and the predominance of proprietary map data in the UK and elsewhere. Since then, it has grown to over two million registered users, which may collect data using manual survey, GPS devices, aerial photography, and other free sources, or use their own local knowledge of the area. Then, these crowdsourced data are made available under the Open Database License. The site is supported by the OpenStreetMap Foundation, which is a non-profit organization registered in England and Wales. Quality assurance and quality control processes are performed to examine the completeness, consistency, accuracy, timeliness, and accessibility of the data hosted in the platform (https://wiki.openstreetmap.org/wiki/Quality_assurance, accessed: 23 October 2021).

The data from OSM can be used in various ways including the production of paper maps and electronic maps (similar to Google Maps, for example), geocoding of address and place names, and route planning. OpenStreetMap data have been favorably compared with proprietary data sources. In this study, we apply OSM data to locate major road networks, which are used to distribute emissions from road traffic.

*Appendix A.6. Global Shipping Lanes*

Shipping lanes are extracted from a publicly available image, retrieved from the map of the world oceans [27]. This is a newly developed spatial proxy to distribute CAMS-REG emissions from the shipping sector along the underlying shipping routes. The spatial analysis of this global map is rather low; thus, only one shipping route could be extracted for the sea area of the Athens domain instead of the actual shipping route network.

**Appendix B. Chemistry Transport Model Description and Setup**

To evaluate the performance of emissions as downscaled with the introduced hybrid approach when applied in an urban-scale CTM, we applied the EPISODE-CityChem [22] model to determine the $NO_2$ and $PM_{2.5}$ concentrations in Hamburg (2016) and Athens (December 2018).

EPISODE-CityChem combines a 3D Eulerian grid model with sub-grid Gaussian dispersion models to resolve pollutant dispersion in proximity of point sources and lines sources. On the Eulerian grid, time-dependent 3D concentration fields of the pollutants are calculated by solving the advection/diffusion equation with terms for chemical reactions, dry deposition and wet deposition, and area emissions. The hourly 2D and 3D fields of meteorological variables and the hourly 2D fields of area emissions are given as input to the model with the spatial resolution of the Eulerian grid. In this study, we applied prognostic meteorological fields from the meteorological component of the coupled meteorological and chemistry transport model TAPM [66]. To drive the meteorological module of TAPM, we applied three-hourly synoptic-scale ECMWF ERA5 reanalysis ensemble means for 2016 and 2018 on a longitude/latitude grid at 0.3-degree grid spacing. The meteorological fields simulated with TAPM have a horizontal resolution of 1 km × 1 km and a vertical resolution of 30 layers with different heights, following the EPISODE-CityChem vertical layer structure.

To account for the background air concentrations, we applied hourly Copernicus Atmospheric Monitoring Services (CAMS) ensemble reanalysis for carbon monoxide (CO), ammonia (NH3), NMVOC, NO, $NO_2$, $O_3$, peroxyl nitrates (PANS), particulate matter ($PM_{10}$, $PM_{2.5}$), and $SO_2$. The CAMS regional ensemble is based on nine state-of-the-art numerical air quality models developed in Europe [67]. The spatial resolution of the regional forecast is 0.1 × 0.1 degrees for the whole of Europe, with nine vertical levels, extending from the surface up to 500 hPa, and the time resolution is one hour. The CAMS forecast concentrations are downloaded and interpolated to the horizontal and vertical resolution of the domain, to be considered at the lateral and vertical borders of the urban domains in EPISODE-CityChem simulations.

In EPISODE-CityChem, emissions from point sources are added onto the Eulerian grid concentration during each time step. Emissions from line sources are added to the grid

concentrations following a procedure described in more detail in Hamer et al. (2019) [68]. The combination of a Eulerian grid model with sub-grid Gaussian dispersion models allows for the calculation of ground-level concentrations near pollution sources with high spatial resolution. Moreover, a simplified street canyon model (SSCM) is part of EPISODE-CityChem for better treatment of pollutant dispersion in street canyons in comparison to models without SSCM [69]. The SSCM module computes concentrations for the receptor points that are located in street canyons. To account for all relevant emission sources in the study domain, emission data containing sector-specific and geo-referenced yearly emission totals created with the developed UrbEm approach are processed with the EPISODE-CityChem interface for emission pre-processing, the Urban Emission Conversion Tool (UECT) [22]. UECT creates hourly varying emission input for point sources, line sources, and area source categories using sector specific temporal profiles and vertical profiles, based on annual totals of emissions. Temporal profiles from the SMOKE-EU model [58] are applied in UECT.

**Table A1.** Model description and setup for the case studies of Athens and Hamburg.

| Name (Version) | EPISODE-CityChem (v1.2r) | |
|---|---|---|
| Short description | A Chemistry Transport Model to enable chemistry/transport simulations of reactive pollutants on the city scale. EPISODE is a Eulerian dispersion model developed at the Norwegian Institute for Air Research (NILU) appropriate for air quality studies at the local scale. The CityChem extension, developed at Helmholtz-Zentrum Geesthacht (HZG) is designed for treating complex atmospheric chemistry in urban areas and improved representation of the near-field dispersion. | |
| Reference(s) | Karl et al., 2019 [22]; Hamer et al., 2019 [68] | |
| Availability | The EPISODE model and the CityChem extension are open-source code subject to the Reciprocal Public License ("RPL") Version 1.5, https://opensource.org/licenses/RPL-1.5 (accessed: 23 October 2021). Zenodo. http://doi.org/10.5281/zenodo.1116173 (accessed: 23 October 2021). | |
| Important mechanisms | Gaseous chemistry: EmChem09-mod, including 70 chemical species, 67 thermal reactions, and 25 photolysis reactions (Karl et al., 2019 [22]). <br> Aerosol treatment: $PM_{2.5}$ and $PM_{10}$ are treated as passive tracers. Dry deposition of particles due to Brownian diffusion, impaction, interception, and gravitational settling, as well as wet scavenging (Simpson et al., 2003 [70]) <br> Street canyon dispersion: Simplified street canyon model (SSCM) based on the Operational Street Pollution Model (OSPM; Berkowicz et al., 1997 [69]) using generic canyon classifications. <br> Gaussian sub-grid dispersion: Line source dispersion (HIWAY2) coupled to SSCM. Point source dispersion by segmented plume model (SEGPLU). <br> Local photochemistry (EP10-Plume; Karl et al., 2019 [22]) is applied in the receptor points of the receptor grid ($100 \times 100$ m$^2$). | |
| Boundary AQ conditions | CAMS reanalysis hourly AQ data (http://www.regional.atmosphere.copernicus.eu, accessed: 23 October 2021) | |
| Air pollution emissions | Anthropogenic emission rates from CAMS-REG-AP v3.1 (Denier van der Gon et al., 2010; Kuenen et al., 2011, 2014 [17,67]) | |
| Meteorological fields | The Air Pollution Model (TAPM) [66], fed by synoptic-scale meteorological reanalysis ensemble means (ECMWF ERA 5). | |
| Outputs | Hourly mean mass concentration values ($\mu$g m$^{-3}$) for $O_3$, $NO$, $NO_2$, $H_2O_2$, $N_2O_5$, $HNO_3$, $SO_2$, H2SO4, $CO$, $PM_{2.5}$, $PM_{10}$, NMVOCs (10 individual species). | |
| Vertical grid | 24 levels (from surface to ca. 3.7 km; first layer is 17.5 m thick). | |
| | Athens | Hamburg |
| Horizontal domain | SW corner 23.4° E, 37.8° N ($45 \times 45$ cells of $1 \times 1$ km$^2$, with an embedded receptor grid $100 \times 100$ m$^2$) | SW corner 53.5° E, 9.9° N ($30 \times 30$ cells of $1 \times 1$ km$^2$, with an embedded receptor grid $100 \times 100$ m$^2$) |
| Simulation period | 1–31 December 2018 | 1 January–31 December 2016 |
| Scenarios | CAMS no proxy: original emissions database, no proxies used for the downscaling <br> UrbEm: high-resolution emissions, based on CAMS, disaggregated through selected proxies | |

**Appendix C. Model Evaluation Local Statistics**

In the statistical analysis of the model performance, the following statistical indicators are used: normalized mean bias (*NMB*), standard deviation (SD), root mean square error (*RMSE*), correlation coefficient (*r*), index of agreement (*IOA*), and the fraction of predictions within a factor of two of observations (FAC2). The overall bias captures the average deviations between the model and observed data and the *NMB* is given by:

$$NMB = \frac{\overline{M} - \overline{O}}{\overline{O}} \tag{A1}$$

where $\overline{M}$ and $\overline{O}$ stand for the averaged model and observation results, respectively. The *RMSE* combines the magnitudes of the errors in predictions for various times into a single measure and is defined as:

$$RMSE = \sqrt{\frac{1}{N} * \sum_{i=1}^{N} (M_i - O_i)^2} \tag{A2}$$

where subscript *i* indicates the time step and *N* indicates the number of observations. *RMSE* is a measure of accuracy to compare prediction errors of different models for particular data and not between datasets, as it is scale-dependent. The correlation coefficient (Pearson *r*) for the temporal correlation is defined as:

$$r = \frac{\sum_{i=1}^{n} (O_i - \overline{O}) \cdot (M_i - \overline{M})}{\sqrt{\sum_{i=1}^{n} (O_i - \overline{O})^2 \cdot \sum_{i=1}^{n} (M - \overline{M})^2}}. \tag{A3}$$

The index of agreement is defined as:

$$IOA = 1 - \frac{\sum_{i=1}^{N} (O_i - M_i)^2}{\sum_{i=1}^{N} (|M_i - \overline{M}| + |O_i - \overline{O}|)^2}. \tag{A4}$$

An *IOA* value close to 1 indicates agreement between modeled and observed data. The fraction of modeled values within a factor of two (FAC2) of the observed values are the fraction of model predictions that satisfy the following:

$$0.5 \leq \frac{M_i}{O_i} \leq 2.0. \tag{A5}$$

For evaluation of modeled values in rural areas, the acceptance criteria is FAC2 $\geq$ 0.5, while in urban areas, it is FAC2 $\geq$ 0.3 [71].

**Table A2.** Evaluation statistics for Hamburg. We aggregated all stations of the same type to groups of stations (type).

| Pollutant | Type | Scenario | *n* | FAC2 | *NMB* | *RMSE* | *r* | *IOA* | Mean Mod | Mean Obs | SD Mod | SD Obs |
|---|---|---|---|---|---|---|---|---|---|---|---|---|
| NO$_2$ | urban background | CAMS no proxy | 78,466 | 0.59 | −0.33 | 15.05 | 0.54 | 0.55 | 14.37 | 21.57 | 11.18 | 15.23 |
| | urban background | UrbEm | 78,466 | 0.64 | −0.12 | 15.44 | 0.51 | 0.54 | 19.95 | 21.57 | 16.10 | 15.23 |
| | urban industrial | CAMS no proxy | 17,362 | 0.75 | −0.14 | 17.57 | 0.39 | 0.50 | 26.76 | 31.04 | 14.06 | 16.66 |
| | urban industrial | UrbEm | 17,362 | 0.74 | −0.05 | 17.90 | 0.43 | 0.49 | 29.54 | 31.04 | 16.63 | 16.66 |

**Table A2.** *Cont.*

| Pollutant | Type | Scenario | n | FAC2 | NMB | RMSE | r | IOA | Mean Mod | Mean Obs | SD Mod | SD Obs |
|---|---|---|---|---|---|---|---|---|---|---|---|---|
| | urban traffic | CAMS no proxy | 34,754 | 0.19 | −0.71 | 47.43 | 0.26 | 0.11 | 15.50 | 54.38 | 11.15 | 27.88 |
| | urban traffic | UrbEm | 34,754 | 0.58 | −0.38 | 32.71 | 0.49 | 0.41 | 33.82 | 54.38 | 20.82 | 27.88 |
| $PM_{2.5}$ | urban background | CAMS no proxy | 8092 | 0.70 | −0.27 | 9.84 | 0.36 | 0.55 | 9.69 | 13.44 | 5.21 | 9.67 |
| | urban background | UrbEm | 8092 | 0.71 | −0.23 | 9.78 | 0.36 | 0.55 | 10.10 | 13.44 | 5.42 | 9.67 |
| | urban industrial | CAMS no proxy | 17,327 | 0.75 | −0.21 | 9.42 | 0.34 | 0.54 | 10.56 | 13.32 | 5.62 | 9.17 |
| | urban industrial | UrbEm | 17,327 | 0.76 | −0.14 | 9.46 | 0.32 | 0.54 | 11.40 | 13.32 | 6.12 | 9.17 |
| | urban traffic | CAMS no proxy | 16,751 | 0.67 | −0.38 | 10.48 | 0.42 | 0.49 | 9.32 | 15.15 | 4.99 | 9.57 |
| | urban traffic | UrbEm | 16,751 | 0.76 | −0.19 | 9.67 | 0.37 | 0.54 | 12.17 | 15.15 | 6.00 | 9.57 |

**Table A3.** Evaluation statistics for Athens. We aggregated all stations of the same type to groups of stations (type).

| Pollutant | Type | Scenario | n | FAC2 | NMB | RMSE | r | IOA | Mean Mod | Mean Obs | SD Mod | SD Obs |
|---|---|---|---|---|---|---|---|---|---|---|---|---|
| $NO_2$ | urban background | CAMS no proxy | 4131 | 0.29 | −0.61 | 22.31 | 0.28 | 0.41 | 8.60 | 22.83 | 9.42 | 17.59 |
| | urban background | UrbEm | 4131 | 0.32 | −0.52 | 21.43 | 0.33 | 0.43 | 10.44 | 22.83 | 12.14 | 17.59 |
| | urban industrial | CAMS no proxy | 1166 | 0.47 | −0.32 | 22.58 | 0.09 | 0.32 | 19.68 | 31.46 | 13.43 | 15.89 |
| | urban industrial | UrbEm | 1166 | 0.73 | −0.04 | 20.27 | 0.43 | 0.45 | 30.71 | 31.46 | 20.91 | 15.89 |
| | urban traffic | CAMS no proxy | 3620 | 0.17 | −0.74 | 39.47 | 0.34 | 0.20 | 10.92 | 42.92 | 7.99 | 24.64 |
| | urban traffic | UrbEm | 3620 | 0.45 | −0.47 | 29.89 | 0.50 | 0.42 | 22.51 | 42.92 | 17.48 | 24.64 |
| $PM_{2.5}$ | urban background | CAMS no proxy | 1352 | 0.40 | −0.53 | 8.33 | −0.01 | 0.09 | 4.92 | 10.82 | 3.45 | 4.83 |
| | urban background | UrbEm | 1352 | 0.36 | −0.59 | 8.97 | −0.04 | 0.00 | 4.22 | 10.82 | 3.74 | 4.83 |
| | urban industrial | CAMS no proxy | 162 | 0.65 | 0.31 | 19.85 | 0.57 | 0.51 | 25.78 | 23.01 | 16.68 | 22.04 |
| | urban industrial | UrbEm | 162 | 0.59 | 0.57 | 26.13 | 0.60 | 0.37 | 29.14 | 23.01 | 21.46 | 22.04 |
| | urban traffic | CAMS no proxy | 1482 | 0.73 | 0.06 | 23.59 | 0.42 | 0.56 | 26.89 | 25.41 | 19.35 | 24.03 |

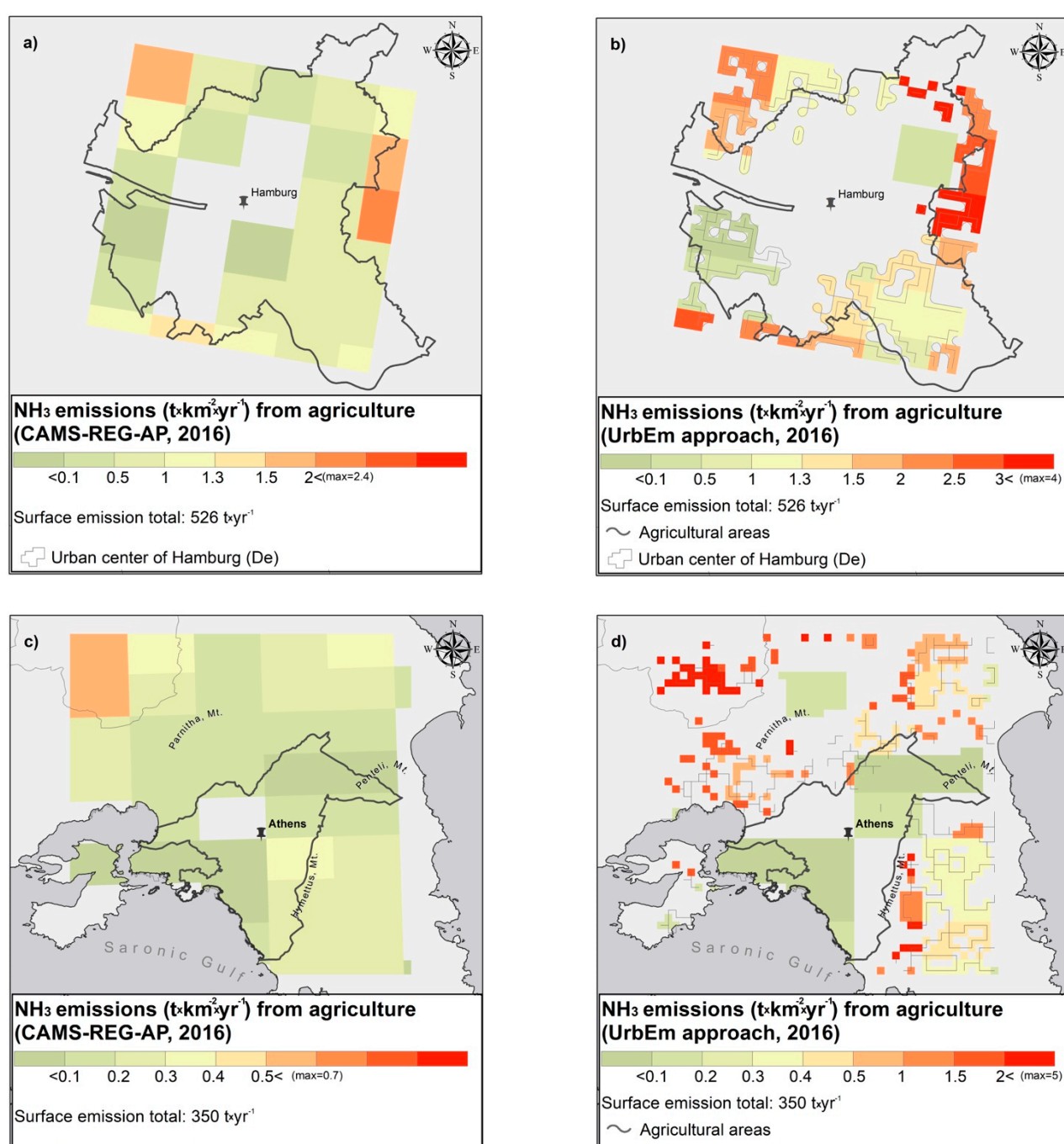

**Figure A1.** NH$_3$ emission fields from agriculture for Hamburg (**a**,**b**) and Athens (**c**,**d**), as originally provided by CAMS (left) and as produced by the UrbEm framework (right). Symbols and isopleths of the main proxies used per source type for the spatial disaggregation of CAMS toward the 1 km by 1 km grid are shown on the maps of the right column (UrbEm outputs).

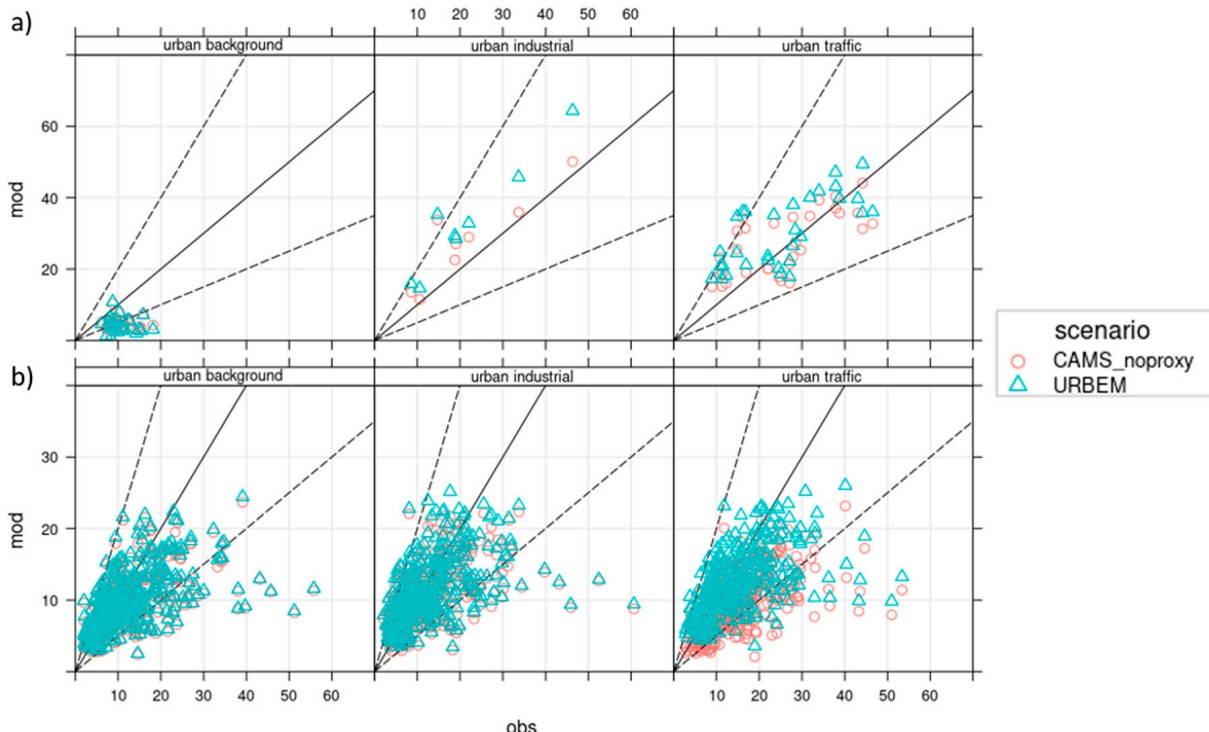

**Figure A2.** Daily PM$_{2.5}$ (µg m$^{-3}$) scatter plots for (**a**) Athens, December 2018 and (**b**) Hamburg, 2016. The observed (obs) versus modeled (mod) mass concentrations are shown, the latter when applying the original CAMS emissions (CAMS no proxy) and high-resolution emissions using the newly developed approach (UrbEm).

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
