# Peer review of "The UrbEm Hybrid Method to Derive High-Resolution Emissions for City-Scale Air Quality Modeling"

_atmosphere, doi:10.3390/atmos12111404_

Round 1
Reviewer 1 Report
In article the UrbEm approach of downscaling gridded emission inventories is developed, investing upon existing, open access and credible emission data sources. The article concerns an important problem of air quality in the city. The overview provided may be useful in further identifying this issue. In this sense, the subject of the reviewed article is important. It also belongs to the scope of the Atmosphere journal. In my opinion, the text is informative and well presented.
The research method has been clearly and comprehensively defined. The research results confirm the possibility of using the UrbEm approach in cities with similar characteristics (in similar case studies).
My remarks concern issues that if developed could support the interpretation of conclusion.
1. (2.2.3 Line sources; regarding the text written in lines: 278-281) How does the presented model analyze the variability of emission sources? Is it possible to model for variable traffic volumes for specific road segments? or for variable parameters for point sources? Such an analysis is important because it allows to determine the air quality in a highly urbanized city in relation to a variable spatial structure. This problem was described, among others, by: https://doi.org/10.3390/rs12182885 or https://doi.org/10.3390/ijerph18137087 et all. In my opinion, this problem should be better described.
2. The conclusions lack critical remarks (for example, in terms of the quality of the spatial data used).
3. In the presented method, the authors adopted some generalizations, e.g. "a main aim of the UrbEm approach is to create a generalized framework that is easy-to-use without such specific data". This generalization raises my doubts. Will this generalization be correct for other locations and conditions? Has it been clearly described? It is worth adding a critical remark to the conclusions.
4. The conclusions lack direction for future research and generalizations.
Author Response
Dear Reviewer, please find attached our answers to your review attached. Due to further changes made in another review, we added this too.

Reviewer 2 Report
This manuscript describes the methodology for the creation of emission inventories for urban air quality simulations applicable to the whole of Europe. The performance of the produced inventories was evaluated by comparing the concentration fields of the different pollutants simulated for the urban scale with local measurements of these pollutants. The demonstration cities were Hamburg and Athens.
This is a very interesting and relevant work, well described, but it also presents opportunities for improvement that should be addressed before being published.
Minor
Among other details:
- I believe that, somewhere after line 55, a reference should be made to the new WHO global air quality guidelines; (https://apps.who.int/iris/handle/10665/345329);
- In line 351, there is an inaccuracy, as the limit value in directive 2008/50/EC for the protection of human health relates to NO2 and not NOx, unless the authors refer to the critical value for the protection of vegetation (30µg.m-3), but I don't think so.
- The legend in Figure 6 should be concern Athens, not Hamburg
- In line 623, in the text relating to Figure 8, NOx is referred to, however the legend of the Figure 8 refers to NO2. Moreover, it is noted throughout the text some confusion between the pollutant that is usually used in emission databases, NOx, and the relevant pollutant in air quality, NO2. This should be clarified. The emission database, as usual, is about NOx, but in air quality NO2 is the pollutant of concern. Does the model calculate NO2 or NOx concentrations? or NO and NOx? In the paper, when the authors refer to concentration in the atmosphere, do they mean NO2 or NOx?
- The information that appears in row 623 regarding the grid size of CAM-REG emissions should appear also in section 2 and not only in section 4;
- In rows 770 and 778, sulphur oxides, should be referred as SOx, not SO2.
Major
In my opinion, the discussion in the section "3.2.2. Comparison of predictions and observations" is very poor considering the amount of information available. Tables AIII-1 and AIII-2 are not discussed in sufficient depth and there is only reference to some statistical indicators in terms of percentage differences without a real discussion on what was obtained and what could be expected.
It is not clear why the simulations improve with UrbEm for NOx, but hardly affect the model performance for PM2.5.
A very relevant aspect is whether runs with the CAMS emission database differ significantly in statistical terms from runs with the UrbEm emission model for the different pollutants. Are the observed differences significant or not?
In my opinion, there is much work to be done in analysing the data obtained.
Author Response
Dear Reviewer, thank you for you review. We addressed all your comments and replied to it in the attached manuscript (Reviewer 2). Due to changes recommended by another review, we also provided our answers to the other review (Reviewer 1). Thanks again for your time and effort to review our manuscript.

Round 2
Reviewer 2 Report
Dear authors,
In my opinion, the manuscript has improved significantly, although I still think the discussion could have gone a bit further.
In any case I think it already deserves publication.
Thank you.